**REPORT**

# DOT1L modulates the senescence-associated secretory phenotype through epigenetic regulation of IL1A

Kelly E. Leon[1,2], Raquel Buj[1], Elizabeth Lesko[3] , Erika S. Dahl[2], Chi-Wei Chen[1] , Naveen Kumar Tangudu[1] , Yuka Imamura-Kawasawa[4], Andrew V. Kossenkov[5] , Ryan P. Hobbs[3], and Katherine M. Aird[1]

Oncogene-induced senescence (OIS) is a stable cell cycle arrest that occurs in normal cells upon oncogene activation. Cells undergoing OIS express a wide variety of secreted factors that affect the senescent microenvironment termed the senescence-associated secretory phenotype (SASP), which is beneficial or detrimental in a context-dependent manner. OIS cells are also characterized by marked epigenetic changes. We globally assessed histone modifications of OIS cells and discovered an increase in the active histone marks H3K79me2/3. The H3K79 methyltransferase disruptor of telomeric silencing 1-like (DOT1L) was necessary and sufficient for increased H3K79me2/3 occupancy at the *IL1A* gene locus, but not other SASP genes, and was downstream of STING. Modulating DOT1L expression did not affect the cell cycle arrest. Together, our studies establish DOT1L as an epigenetic regulator of the SASP, whose expression is uncoupled from the senescence-associated cell cycle arrest, providing a potential strategy to inhibit the negative side effects of senescence while maintaining the beneficial inhibition of proliferation.

## Introduction

Cellular senescence is defined as a stable cell cycle arrest that can occur due to multiple stimuli, such as oncogenic stress (Serrano et al., 1997). Although the induction of senescence upon oncogene activation (termed oncogene-induced senescence [OIS]) can result in tumor suppression, it may also result in tumor promotion and progression (Coppé et al., 2006; Ritschka et al., 2017; Sparmann and Bar-Sagi, 2004). One hallmark of senescence is the senescence-associated secretory phenotype (SASP), a proinflammatory microenvironment composed of cytokines, chemokines, matrix metalloproteinases, and other secreted factors (Acosta et al., 2008; Coppé et al., 2008; Kuilman et al., 2008). While the SASP may enhance immune cell recruitment and clearance of senescent cells, it also has detrimental side effects, resulting in chronic inflammation that contributes to tumorigenesis and chemoresistance (Coppé et al., 2008). Therefore, further understanding of how to restrict the negative effects of the SASP while maintaining the senescence-associated cell cycle arrest has implications in transformation and tumorigenesis.

Previous studies have demonstrated that the proinflammatory cytokines and chemokines of the SASP are transcriptionally regulated by nuclear factor-κB (NF-κB) and CCAAT/enhancer-binding protein β (C/EBPβ; Acosta et al., 2008; Kuilman et al., 2008). One key component of the SASP is interleukin-1α (IL1A), which is thought to be one of the critical upstream regulators of other SASP-related genes (Gardner et al., 2015; Ong et al., 2018; Orjalo et al., 2009; Wiggins et al., 2019). Indeed, cell surface IL1A expression is necessary for a positive feedback loop to promote transcription of multiple cytokines, such as *IL6*, *CXCL8* (encoding IL8), and *IL1B* (Gardner et al., 2015; Lau et al., 2019; Orjalo et al., 2009). While target of rapamycin has been implicated in translational regulation of *IL1A* (Laberge et al., 2015), less is clear about its transcriptional regulation, especially since it seems to be in part upstream of NF-κB (Orjalo et al., 2009). Furthermore, recent studies have demonstrated that the innate DNA-sensing pathway cyclic GMP-AMP synthase (cGAS)-stimulator of interferon genes (STING) is an upstream regulator of the SASP (Glück et al., 2017; Yang et al., 2017). Increased DNA damage caused by OIS and decreased nuclear lamin expression results in cytoplasmic chromatin fragments that activate cGAS-STING and the downstream effectors interferon regulator factor 3 (IRF3) and NF-κB (Di Micco et al., 2006;

.....................................................................................................................................................................................................

[1]Department of Pharmacology and Chemical Biology, University of Pittsburgh Medical Center Hillman Cancer Center, University of Pittsburgh School of Medicine, Pittsburgh, PA; [2]Biomedical Sciences Graduate Program, Penn State College of Medicine, Hershey, PA; [3]Department of Dermatology, Penn State College of Medicine, Hershey, PA; [4]Department of Pharmacology, Penn State College of Medicine, Hershey, PA; [5]The Wistar Institute, Philadelphia, PA.

Correspondence to Katherine M. Aird: kaa140@pitt.edu.

Dunphy et al., 2018; Glück et al., 2017; Mackenzie et al., 2017). Although cGAS-STING has been implicated in regulating the SASP during OIS, whether and how cGAS-STING affects the transcription of the key SASP regulator *IL1A* is unknown.

In addition to the SASP, another hallmark of senescence is a marked change in histone modifications (Chandra et al., 2012; Narita et al., 2003; Zhang et al., 2007). Di- and trimethylation of H3K9, repressive histone marks that are found in heterochromatin, are known to decrease proliferation-promoting genes during OIS (Narita et al., 2003). However, SASP genes are protected from heterochromatinization via HMGB2 (Aird et al., 2016), allowing their continued and increased transcription. Previous reports have demonstrated that active and repressive histone marks, such as H3K4me3 and H3K27me3, respectively, affect multiple senescence phenotypes, including the SASP (Capell et al., 2016; Ito et al., 2018; Shah et al., 2013). Another histone mark that may have a role in senescence is H3K79 (Kim et al., 2012; Wang et al., 2010), which is associated with active transcription (Wood et al., 2018). Disruptor of telomeric silencing 1-like (DOT1L) is the sole methyltransferase for H3K79 (Feng et al., 2002). Methylation of H3K79 by DOT1L has been implicated in contributing to the DNA damage response (DDR; Kari et al., 2019). Additionally, previous studies have linked decreased DOT1L and H3K79 methylation to cell cycle inhibition and senescence (Barry et al., 2009; Kim et al., 2012). However, whether DOT1L or the active histone mark H3K79 regulate the SASP is unknown.

Here, we found that the active marks H3K79me2/3 were enriched at the *IL1A* gene locus in OIS cells. Mechanistically, we determined that the H3K79 methyltransferase DOT1L is upregulated in OIS, and DOT1L is both necessary and sufficient for H3K79me2/3 occupancy at the *IL1A* locus, contributing to subsequent expression of downstream SASP genes without altering other senescence phenotypes. This phenotype correlated with changes in C/EBPβ expression but not changes in p65 phosphorylation. Upregulation of DOT1L required STING, suggesting that DOT1L upregulation downstream of STING is a feed-forward loop to increase SASP gene expression in part via C/EBPβ. Altogether, we determined that DOT1L is a mediator of IL1A and SASP expression, which is uncoupled from the senescence-associated cell cycle arrest.

## Results and discussion

### H3K79 di- and trimethylation marks are increased at the *IL1A* locus, which corresponds to increased expression of the methyltransferase DOT1L

To globally determine changes in histone modifications during senescence, we performed unbiased epiproteomics in the classic model of OIS: HRAS^G12V overexpression in IMR90 cells (Serrano et al., 1997), hereafter referred to as RAS (Fig. 1, A–E; and Fig. S1 A). Multiple histone marks were significantly altered, including H3K27me3 and H3K36me1, which have been previously published (Fig. 1 F and Table S1; Chandra et al., 2012; Ito et al., 2018). Interestingly, H3K79me2 and H3K79me3, active histone marks, were increased during RAS-induced senescence, while the unmodified form of H3K79 was decreased (Fig. 1 G). Increased

H3K79me2/3 were also observed by Western blotting (Fig. 1 H). H3K79me2/3 were also increased in BJ-hTERT cells that were induced to senescence through BRAF^V600E overexpression (hereafter referred to as BRAF), suggesting that this is not oncogene or cell line specific (Fig. S1, B–G). Similar to our observations, others have observed increased H3K79 methylation during OIS (Chicas et al., 2012) and increased H3K79 methylation at telomeres in yeast-aging models (Rhie et al., 2013). We therefore sought to understand the functional role of H3K79 methylation in the regulation of OIS. To determine the global occupancy of H3K79me3 during OIS, we performed chromatin immunoprecipitation followed by next-generation sequencing (ChIP-seq) on RAS-induced senescent cells (Gene Expression Omnibus [GEO] accession no. GSE156591). As H3K79 is an active histone mark associated with transcriptional upregulation (Wood et al., 2018), we cross compared ChIP-seq peaks with genes that increase with a fold change (FC) >1.5 and q-value (false discovery rate [FDR]) <0.25 in our RNA sequencing (RNA-seq) analysis (GEO accession no. GSE156648). This resulted in 95 common genes that were enriched in a number of pathways related to an inflammatory response (Fig. 1 I and Table S2). Therefore, we reasoned that H3K79me3 may regulate the SASP (Chien et al., 2011; Kuilman et al., 2008; Rodier et al., 2009). Upon further analysis of our ChIP-seq data, we discovered that H3K79me3 occupancy was enriched at the *IL1A* locus (Fig. 1 J). Although we did observe an increase in *IL6*, *IL1B*, and *CXCL8* expression (Fig. 1 K), which are known to be transcriptionally regulated (Aird et al., 2016; Buj et al., 2021; Capell et al., 2016; Orjalo et al., 2009), H3K79me3 occupancy was not enriched at these loci (Fig. 1 L). As IL1A is a critical upstream regulator of the SASP (Gardner et al., 2015; Laberge et al., 2015; Ong et al., 2018; Orjalo et al., 2009; Wiggins et al., 2019), these data suggest that H3K79me3 may be important for initiating the SASP via *IL1A* transcription. H3K79me2 and H3K79me3 have distinct histone patterns, with H3K79me2 preferentially at the promoter region and H3K79me3 within the gene body (Fig. 1 M; Guenther et al., 2007). ChIP-quantitative PCR (qPCR)–validated increased H3K79me2 and H3K79me3 occupancy at the *IL1A* locus in both OIS models (Fig. 1, N and O; and Fig. S1 H). Increased H3K79me2/3 at the *IL1A* gene locus corresponded to an increase in its transcription and expression at the cell surface (Fig. 1, P and Q; and Fig. S1 I), which may drive the downstream SASP. Together, these data suggest that H3K79me2/3 may play a potential role in SASP gene expression during senescence via transcriptional activation of *IL1A*.

Next, we aimed to determine the mechanism of increased H3K79 methylation in OIS. DOT1L is the only methyltransferase for H3K79 (Feng et al., 2002; Lacoste et al., 2002), while the lysine demethylases KDM2B and KDM4D have been implicated as H3K79 demethylases (Jbara et al., 2017; Kang et al., 2018; Leon and Aird, 2019). RNA-seq analysis of all three genes indicated that *DOT1L* is significantly upregulated in RAS-induced senescent cells, while there is no difference in either *KDM2B* or *KDM4D* expression (Fig. 1, R and S; and Fig. S1 J). *DOT1L* was also upregulated in melanocytes and fibroblasts induced to senesce by BRAF (Fig. S1, K–M; Pawlikowski et al., 2013). Consistent with mRNA expression, we observed an increase in DOT1L protein

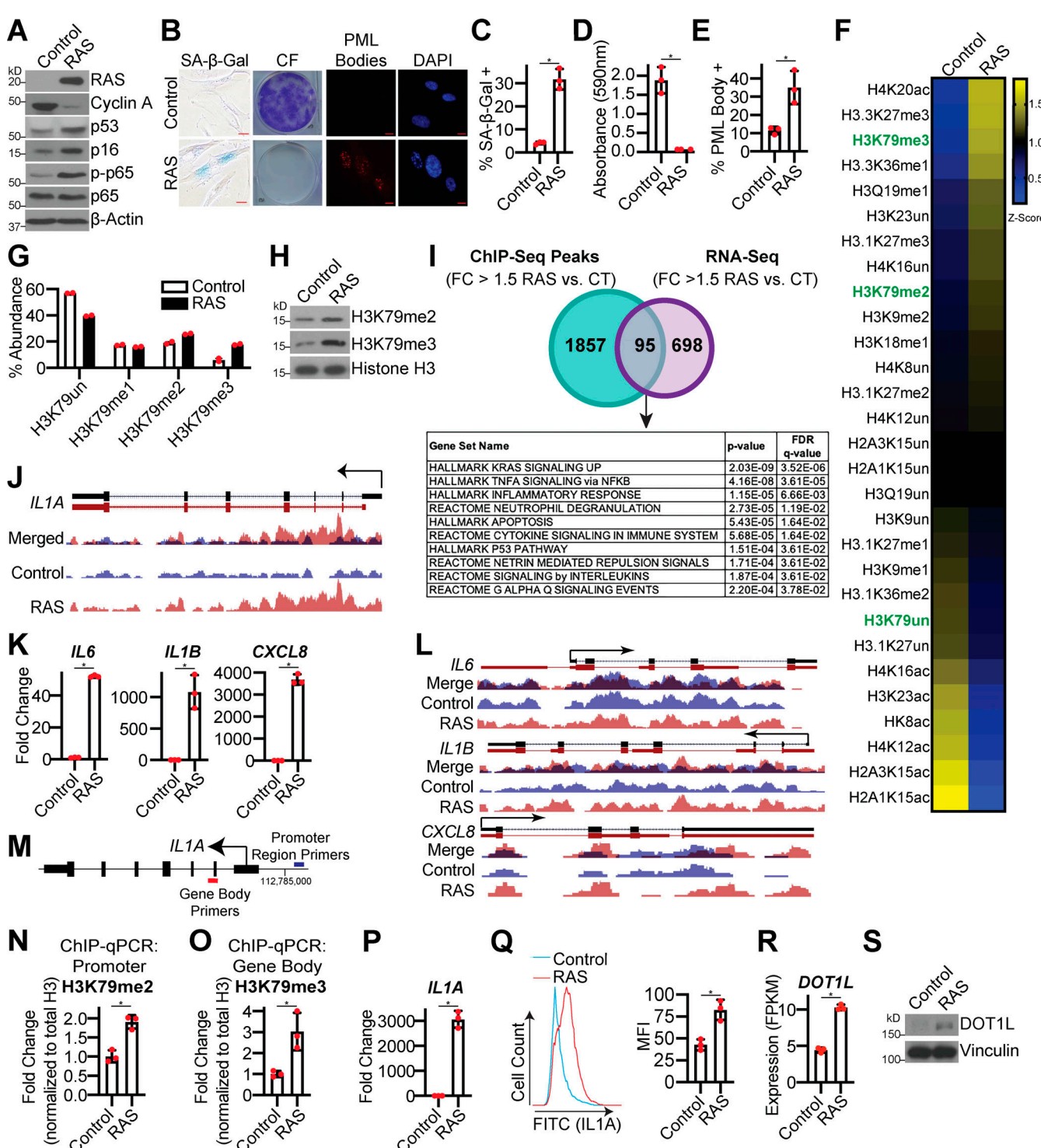

Figure 1. **H3K79me2/3 are increased at the IL1A locus during OIS, corresponding to increased DOT1L methyltransferase expression.** IMR90 cells were infected with retrovirus expressing HRas^G12V (RAS) or empty vector control. See Fig. S1 for time points. **(A)** Immunoblot analysis of the indicated proteins. β-Actin was used as a loading control. One of three independent experimental replicates is shown. **(B)** Senescence-associated β-galactosidase (SA-β-Gal) activity, colony formation (CF), and promyelocytic leukemia (PML) body immunofluorescence. Shown are representative images from one of three independent experimental replicates. Scale bar, 10 μm. **(C)** Quantification of SA-β-Gal activity in B. One of three independent experimental replicates is shown. Data represent mean ± SD ($n$ = 3, where each dot represents >100 cells counted). *, P < 0.0004 by Student's $t$ test. **(D)** Quantification of CF in B. One of three independent experimental replicates is shown. Data represent mean ± SD ($n$ = 3). *, P < 0.0008 by Student's $t$ test. **(E)** Quantification of PML body immunofluorescence in B. One of three independent experimental replicates is shown. Data represent mean ± SD ($n$ = 3, where each dot represents >200 cells counted). *, P < 0.0118 by Student's $t$ test. **(F)** Heat map of unmodified, methylated, and acetylated histones determined using LC-MS/MS. Only modifications with FDR <0.1 are shown. H3K79 is marked in green. See Table S1 for raw data. **(G)** H3K79un (unmodified), H3K79me1, H3K79me2, and H3K79me3 percent abundance was determined by LC-MS/MS. Data represent mean ± SEM ($n$ = 2). **(H)** H3K79me2 and H3K79me3 immunoblot analysis of chromatin fraction. Total

histone H3 was used as a loading control. One of three independent experimental replicates is shown. **(I)** Cross-referencing of H3K79me3 ChIP-seq data and RNA-seq data. A total of 1,952 genes showed increased H3K79me3 occupancy (FC >1.5 RAS versus control; Table S5), and a total of 793 genes were significantly increased by RNA-seq (FC >1.5 RAS versus control; q-value [FDR] <0.25; Table S6). The 95 genes (listed in Table S2) that overlapped were subjected to GSEA. **(J)** H3K79me3 ChIP-seq track at the *IL1A* gene locus. Blue indicates H3K79me3 binding in control cells, whereas red indicates H3K79me3 binding in RAS cells. **(K)** *IL6*, *IL1B*, and *CXCL8* mRNA expression. One of five independent experimental replicates is shown. Data represent mean ± SD (n = 3). *, P < 0.003 by Student's *t* test. **(L)** H3K79me3 ChIP-seq track at the *IL6*, *IL1B*, and *CXCL8* gene loci. Blue indicates binding of H3K79me3 in control cells, whereas red indicates binding of H3K79me3 binding in RAS cells. **(M)** Schematic of ChIP-qPCR primers for *IL1A* promoter region (H3K79me2 enrichment, indicated in blue) and gene body (H3K79me3 enrichment, indicated in red). **(N and O)** H3K79me2 binding to the *IL1A* promoter region (N) and H3K79me3 binding to the gene body (O) was determined by ChIP-qPCR and normalized to total histone H3 binding at the same site. One of three independent experimental replicates is shown. Data represent mean ± SD (n = 3). *, P < 0.05 by Student's *t* test. **(P)** *IL1A* mRNA expression. One of five independent experimental replicates is shown (n = 3). Data represent mean ± SD (n = 3). *, P < 0.001by Student's *t* test. **(Q)** Cell surface–bound IL1A was determined by flow cytometry. One of four independent experimental replicates is shown. Data represent mean ± SD (n = 3). *, P < 0.01 by Student's *t* test. **(R)** *DOT1L* mRNA expression from RNA-seq. Three technical replicates from one experiment are shown. Data represent mean ± SD (n = 3). *, P < 0.0001 by Student's *t* test. **(S)** Immunoblot analysis of DOT1L. Vinculin was used as a loading control. One of three independent experimental replicates is shown. MFI, median fluorescence intensity.

expression in both RAS- and BRAF-induced senescent cells (Fig. 1 S and Fig. S1 N). Upregulation of DOT1L expression in all these models correlated with increased *IL1A* (Fig. 1 P; and Fig. S1, I, L, and M). Consistently, we also observed a positive correlation between *DOT1L* and *IL1A* expression in mouse papillomas treated with 7,12-dimethylbenz[a]anthracene (DMBA)/12-O-tetradecanoylphorbol-13-acetate (TPA; Fig. S1 O), a known inducer of OIS and the SASP (Alimirah et al., 2020; Ritschka et al., 2017), suggesting that this phenomenon also occurs in vivo. These data provide evidence to demonstrate that DOT1L upregulation upon OIS induction occurs in multiple models and cell types and is associated with increased *IL1A*. Indeed, previous reports have shown that DOT1L suppresses senescence in yeast (Kozak et al., 2010) and plays a protective role in UV-induced melanomagenesis (Zhu et al., 2018), although other studies have shown the opposite (Barry et al., 2009; Jones et al., 2008; Kim et al., 2012; Nassa et al., 2019; Roidl et al., 2016; Song et al., 2020; Yang et al., 2019; Zhang et al., 2014), suggesting that DOT1L activity and 1H3K79 methylation are context dependent. The discrepancy between these studies is not clear; it is possible that this is an inherent distinction between model systems. At least in the context of OIS versus replicative senescence in human cells, it is possible that oncogenic stress, which is generally more robust and associated with a hyperproliferative phase that leads to replication stress and DNA damage (Aird et al., 2013; Di Micco et al., 2006), specifically leads to DOT1L upregulation and H3K79 methylation. Although further studies are required to investigate the context dependency of DOT1L and H3K79 methylation, our data demonstrate that DOT1L activity is increased in OIS.

## DOT1L expression is necessary for SASP gene expression

We next aimed to confirm that DOT1L upregulation is necessary for H3K79 methylation and *IL1A* expression in OIS cells. Knockdown of DOT1L decreased H3K79me2/3 in RAS-induced senescent cells and decreased occupancy of DOT1L and H3K79me2/3 at the *IL1A* locus but not at other SASP loci (Fig. 2, A–C). Interestingly, we noted an overall decrease in H3K79me2/3 occupancy at the *IL6*, *IL1B*, and *CXCL8* loci in RAS-induced senescent cells, which may further point to the specificity of DOT1L-mediated regulation of *IL1A*. Knockdown of DOT1L during RAS-induced senescence also corresponded to decreased *IL1A* mRNA expression, IL1A expression at the cell membrane, and

decreased transcription and secretion of downstream SASP factors (Fig. 2, D–G and Table S3). The decrease in SASP was not due to rescue of the senescence-associated cell cycle arrest (Fig. 2, H–L; and Fig. S2 A). DOT1L is also a regulator of DDR, and H3K79 methylation promotes 53BP1 binding to sites of DNA double-strand breaks (FitzGerald et al., 2011; Huyen et al., 2004). Interestingly, we did not observe marked changes in 53BP1 or γH2AX foci upon DOT1L knockdown (Fig. 2, H and M), suggesting that DOT1L regulates the SASP outside its role in the DDR. Knockdown of DOT1L in BRAF-induced senescent cells or pharmacological inhibition of DOT1L in RAS-induced senescent cells decreased H3K79 methylation and SASP expression while maintaining the senescence-associated cell cycle arrest (Fig. 2, N–R-; and Fig. S1, B–F). These data indicate that DOT1L expression is necessary for the SASP but is dispensable for other senescent cell phenotypes.

We next aimed to determine whether the observed changes in DOT1L-mediated *IL1A* expression are directly linked to downstream SASP transcription and the effects of DOT1L knockdown on SASP transcription factors p65 (NF-κB subunit) and C/EBPβ (Chien et al., 2011; Kuilman et al., 2008; Orjalo et al., 2009). Overexpression of IL1A in senescent cells with DOT1L knockdown rescued expression of *IL6*, *IL1B*, and *CXCL8* (Fig. 3, A and B), consistent with the idea that DOT1L regulates the SASP directly through IL1A. Interestingly, we did not observe changes in phosphorylated p65 (p-p65; Fig. 3 C), although one possible caveat to our study is that the nuclear translocation (Fagerlund et al., 2005; Liang et al., 2013) or DNA-binding ability (Gerritsen et al., 1997; Perkins, 2007) of p65 may be affected by DOT1L knockdown. In contrast, C/EBPβ expression was decreased by DOT1L knockdown and partially rescued by IL1A overexpression (Fig. 3, C and D). Consistently, previous reports have shown that knockdown or activation of IL1A positively correlates with C/EBPβ expression (Hop et al., 2019; Montes et al., 2015; Yang et al., 2015). Similar to what others have shown, our data suggest that changes in C/EBPβ and *IL6*, *IL1B*, and *CXCL8* expression are downstream of IL1A (Orjalo et al., 2009) and not simply a consequence of decreased C/EBPβ due to DOT1L knockdown. Indeed, *CEBPB* is likely not a direct target of DOT1L as H3K79me3 was not enriched at this locus (Fig. 3 E). Altogether, our data directly demonstrate that DOT1L-mediated *IL1A* expression contributes to downstream SASP gene expression and suggest a role for C/EBPβ in this process.

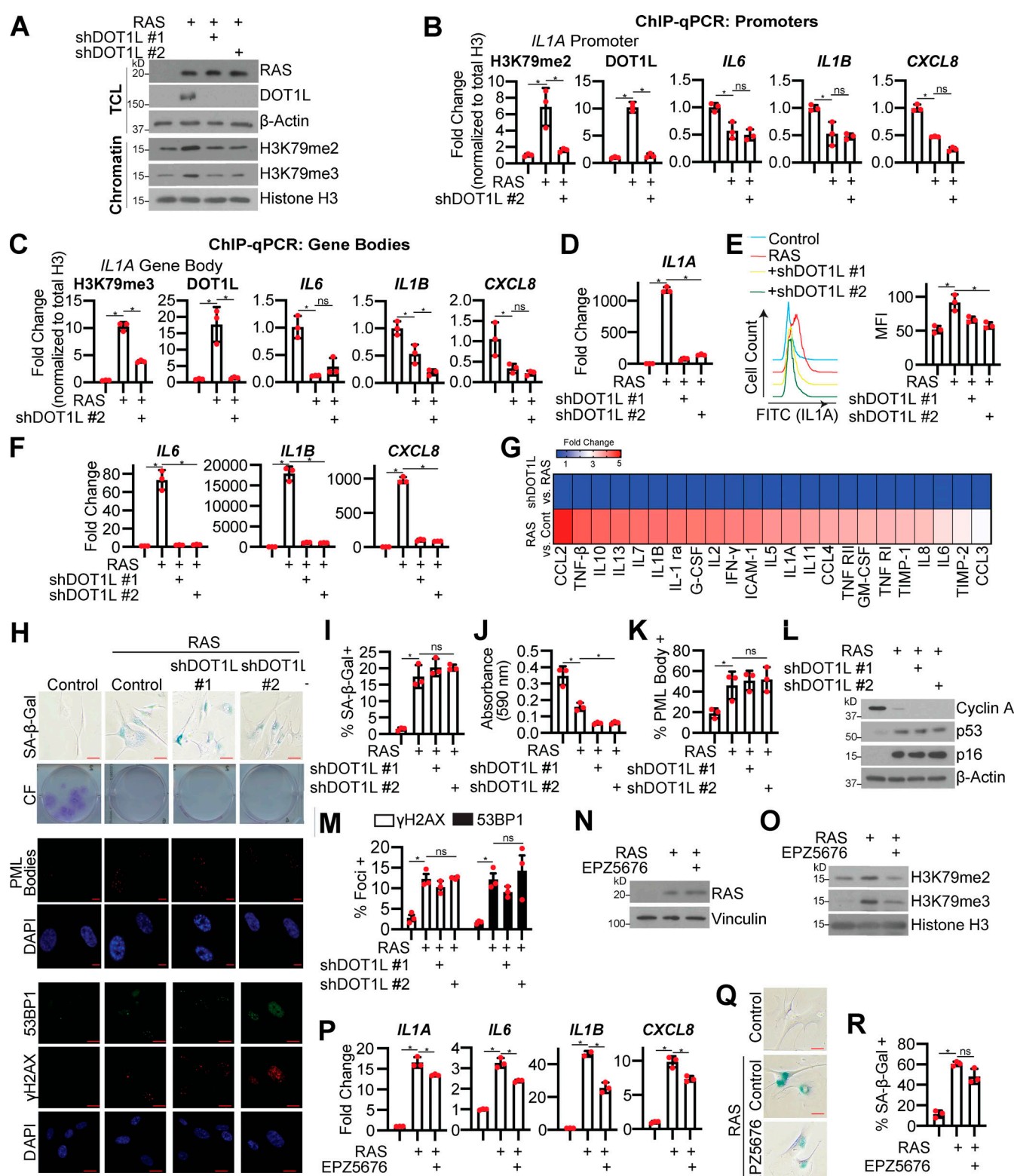

Figure 2. **DOT1L is necessary for H3K79me2/3 at the *IL1A* locus and SASP expression but dispensable for other senescence phenotypes.** IMR90 cells were infected with retrovirus-expressing HRAS^{G12V} (RAS) or empty vector control with or without shRNA to human DOT1L (shDOT1L) or an shGFP control. Details on time points are in Fig. S1 A. **(A)** Immunoblot analysis of total cell lysates (TCL) and chromatin fractions of the indicated proteins. β-Actin was used as a loading control for TCL. Histone H3 was used as a loading control for chromatin fractions. One of three independent experimental replicates is shown. **(B)** H3K79me2 and DOT1L binding to the *IL1A* promoter region and H3K79me2 at the promoters of *IL6*, *CXCL8*, and *IL1B* was determined by ChIP-qPCR and normalized to total histone H3 binding at the same site. One of three independent experimental replicates is shown. Data represent mean ± SD (*n* = 3). *, P < 0.05 by one-way ANOVA with Tukey's multiple comparisons. **(C)** H3K79me3 and DOT1L binding to the *IL1A* gene body and H3K79me3 at the gene bodies of *IL6*, *CXCL8*, and *IL1B* was determined by ChIP-qPCR and normalized to total histone H3 binding at the same site. One of three independent experimental replicates is

shown. Data represent mean ± SD (*n* = 3). *, P < 0.05 by one-way ANOVA with Tukey's multiple comparisons. **(D)** *IL1A* mRNA expression was determined by RT-qPCR. One of five independent experimental replicates is shown. Data represent mean ± SD (*n* = 3). *, P < 0.01 by one-way ANOVA with Tukey's multiple comparisons. **(E)** Cell surface–bound IL1A was determined by flow cytometry. One of four independent experimental replicates is shown. Data represent mean ± SD (*n* = 3). *, P < 0.01 by one-way ANOVA with Tukey's multiple comparisons. **(F)** *IL6*, *IL1B*, and *CXCL8* mRNA expression was determined by RT-qPCR. One of five independent experimental replicates is shown. Data represent mean ± SD (*n* = 3). *, P < 0.01 by one-way ANOVA with Tukey's multiple comparisons. **(G)** Secretion of SASP-related factors was detected using an antibody array. Heat map indicates FC. Data are generated from technical replicates of one independent experiment. Raw data can be found in Table S3. **(H)** Senescence-associated β-galactosidase (SA-β-Gal) activity, colony formation (CF), pro-myelocytic leukemia (PML) body immunofluorescence, γH2AX, and 53BP1 foci. Shown are representative images from one of three independent experimental replicates. Scale bar, 10 µm. **(I)** Quantification of SA-β-Gal activity in H. One of three independent experimental replicates is shown. Data represent mean ± SD (*n* = 3, where each dot represents >100 cells counted). *, P < 0.05 by one-way ANOVA with Tukey's multiple comparisons. **(J)** Quantification of CF in H. One of three independent experimental replicates is shown. Data represent mean ± SD (*n* = 3). *, P < 0.05 by one-way ANOVA with Tukey's multiple comparisons. **(K)** Quantification of PML body immunofluorescence in H. One of three independent experimental replicates is shown. Data represent mean ± SD (*n* = 3, where each dot represents >200 cells counted). *, P < 0.02 by one-way ANOVA with Tukey's multiple comparisons. **(L)** Immunoblot analysis of indicated proteins. β-Actin was used as a loading control. One of three independent experimental replicates is shown. **(M)** Quantification of γH2AX and 53BP1 foci in H. One of three independent experimental replicates is shown. Data represent mean ± SD (*n* = 3, where each dot represents >200 cells counted). *, P < 0.05 by one-way ANOVA with Tukey's multiple comparisons. **(N–R)** IMR90 cells were infected with retrovirus-expressing HRAS$^{G12V}$ (RAS) or empty vector control. Four days after retroviral infection, cells were treated with 1 µM DOT1L inhibitor EPZ5676. **(N)** RAS immunoblot analysis. Vinculin was used as loading control. One of three independent experimental replicates is shown. **(O)** H3K79me2 and H3K79me3 immunoblot analysis was performed on chromatin fractions. Total histone H3 was used as loading control. One of three independent experimental replicates is shown. **(P)** *IL1A, IL6, IL1B*, and *CXCL8* mRNA expression was determined by RT-qPCR. One of three independent experimental replicates is shown. Data represent mean ± SD (*n* = 3). *, P < 0.01 by one-way ANOVA with Tukey's multiple comparisons. **(Q)** SA-β-Gal activity. Shown are representative images from one of three independent experimental replicates. Scale bar, 10 µm. **(R)** Quantification of SA-β-Gal activity in P. One of three independent experimental replicates is shown. Data represent mean ± SD (*n* = 3, where each dot represents >100 cells counted). *, P < 0.0005 by one-way ANOVA with Tukey's multiple comparisons. Cont, control; MFI, median fluorescence intensity.

## DOT1L is sufficient to modestly increase SASP expression

We next sought to determine whether DOT1L and H3K79me2/3 are sufficient for SASP induction. Toward this goal, we overexpressed DOT1L in normal IMR90 or BJ-hTERT fibroblasts (Fig. 4, A and B; and Fig. S3 A). Overexpression of DOT1L increased H3K79me2/3 and increased occupancy of both DOT1L and these active marks at the *IL1A* locus but not at the *IL6*, *IL1B*, or *CXCL8* loci (Fig. 4, C–E; and Fig. S3 B). Increased DOT1L

corresponded to an increase in *IL1A* transcription, cell surface expression, and downstream SASP gene expression and secretion (Fig. 4, F–I; Fig. S3, C and D; and Table S3). Notably, the increase in SASP gene expression is 10–100 times less in DOT1L-overexpressing cells compared with OIS cells, indicating that other mechanisms are in play to fully promote SASP gene transcription. Overexpression of DOT1L did not affect other markers of senescence (Fig. 4, J–M; and Fig. S3, E–G), suggesting

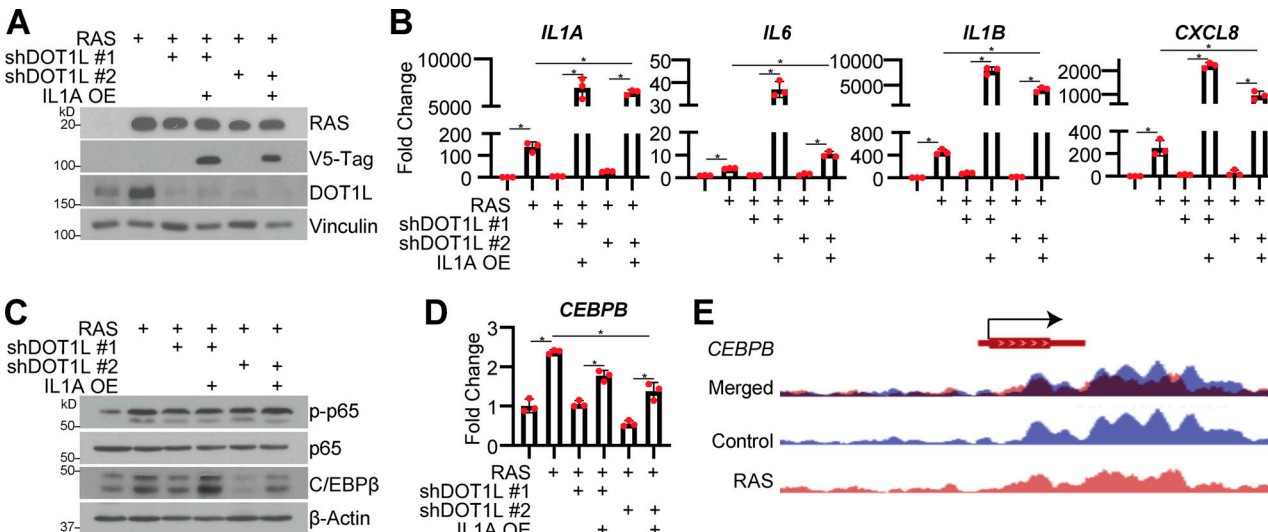

Figure 3. **Overexpression (OE) of IL1A in DOT1L knockdown cells rescues SASP gene expression and corresponds to C/EBPβ expression.** IMR90 cells were infected with retrovirus-expressing HRAS$^{G12V}$ (RAS) or empty vector control followed by a simultaneous infection with or without shRNA to human DOT1L (shDOT1L) or shGFP control and lentivirus-expressing V5-tagged IL1A or empty vector control. **(A)** Immunoblot analysis of the indicated proteins. Vinculin was used as a loading control. One of three independent experimental replicates is shown. **(B)** *IL1A, IL6, IL1B*, and *CXCL8* mRNA expression was determined by RT-qPCR. One of five independent experimental replicates is shown. Data represent mean ± SD (*n* = 3). *, P < 0.01 by one-way ANOVA with Tukey's multiple comparisons. **(C)** Immunoblot analysis of indicated proteins. β-Actin was used as a loading control. One of three independent experimental replicates is shown. **(D)** *CEBPB* mRNA expression was determined by RT-qPCR. One of five independent experimental replicates is shown. Data represent mean ± SD (*n* = 3). *, P < 0.01 by one-way ANOVA with Tukey's multiple comparisons. **(E)** H3K79me3 ChIP-seq track at the *CEBPB* gene locus. Blue indicates binding of H3K79me3 in control cells, whereas red indicates binding of H3K79me3 binding in RAS cells.

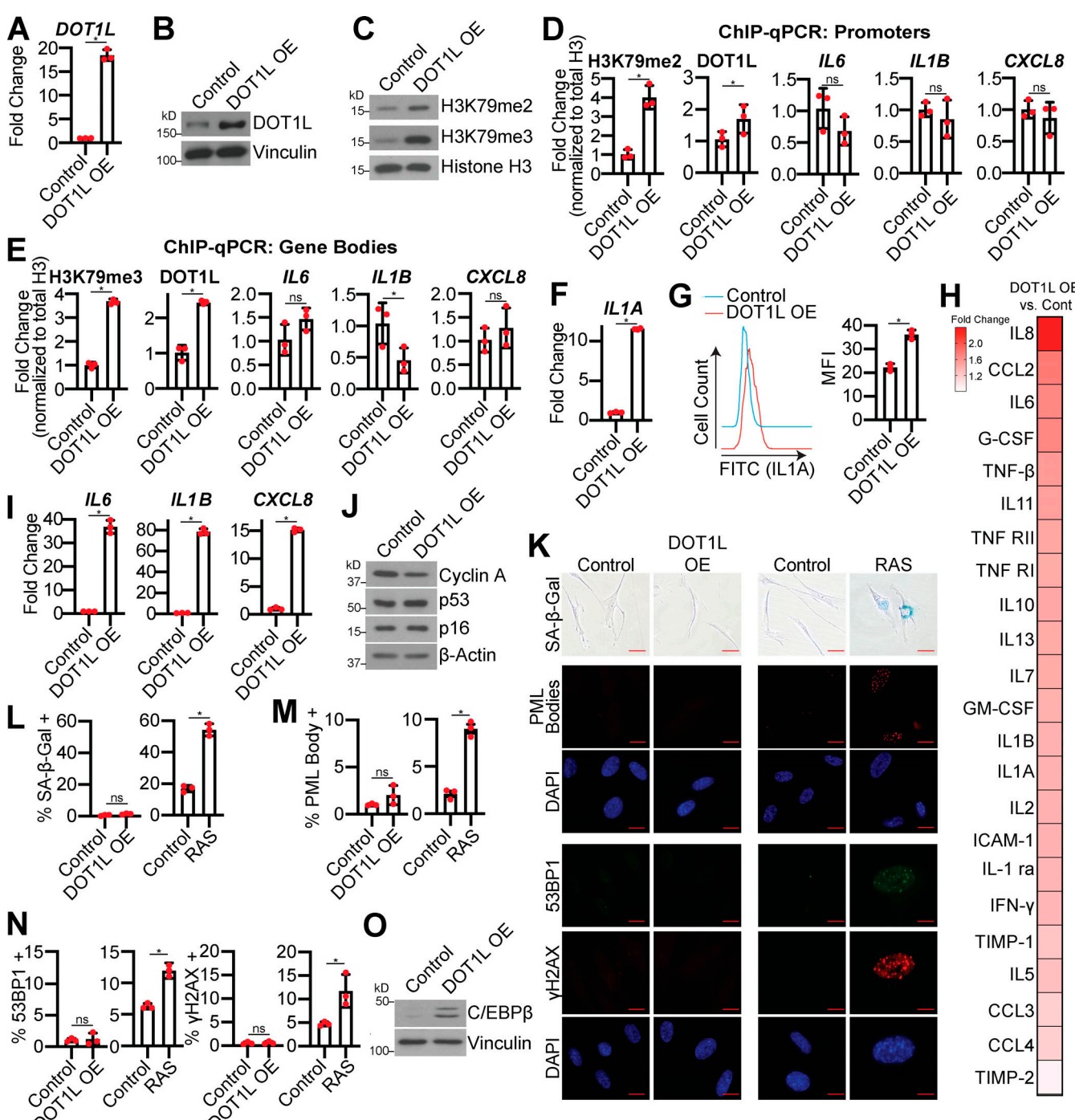

Figure 4. **DOT1L overexpression (OE) increases H3K79me2/3 at the _IL1A_ locus and is sufficient for SASP gene expression but does not affect other senescence phenotypes.** IMR90 cells were infected with retrovirus-expressing human DOT1L or empty vector control. In some experiments, as a positive control, IMR90 cells were infected with retrovirus-expressing HRAS[G12V] (RAS) or empty vector control. **(A)** _DOT1L_ mRNA expression. One of three independent experimental replicates is shown. Data represent mean ± SD (_n_ = 3). *, P < 0.001 by Student's _t_ test. **(B)** DOT1L immunoblot analysis. Vinculin was used as a loading control. One of five independent experimental replicates is shown. **(C)** H3K79me2 and H3K79me3 immunoblot analysis on chromatin fractions. Total histone H3 was used as loading control. One of three independent experimental replicates is shown. **(D)** H3K79me2 and DOT1L binding to the _IL1A_ promoter region and H3K79me2 at the promoters of _IL6_, _CXCL8_, and _IL1B_ was determined by ChIP-qPCR and normalized to total histone H3 binding at the same site. One of three independent experimental replicates is shown. Data represent mean ± SD (_n_ = 3). *, P < 0.05 by Student's _t_ test. **(E)** H3K79me3 and DOT1L binding to the _IL1A_ gene body and H3K79me3 at the gene bodies of _IL6_, _CXCL8_, and _IL1B_ was determined by ChIP-qPCR and normalized to total histone H3 binding at the same site. One of three independent experimental replicates is shown. Data represent mean ± SD (_n_ = 3). *, P < 0.05 by Student's _t_ test. **(F)** _IL1A_ mRNA expression was determined by RT-qPCR. One of five independent experimental replicates is shown. Data represent mean ± SD (_n_ = 3). *, P < 0.006 by Student's _t_ test. **(G)** Cell surface–bound IL1A was determined by flow cytometry. One of four independent experimental replicates is shown. Data represent mean ± SD (_n_ = 3). *, P < 0.001 by Student's _t_ test. **(H)** Secretion of SASP-related factors were detected by antibody array. Heat map indicates FC of DOT1L OE versus control. Data are generated from technical replicates of one independent experiment. Raw data can be found in Table S3. **(I)** _IL6_, _IL1B_, and _CXCL8_ mRNA

expression was determined by RT-qPCR. One of five independent experimental replicates is shown. Data represent mean ± SD ($n = 3$). *, P < 0.0001 by Student's $t$ test. **(J)** Immunoblot analysis of the indicated proteins. β-Actin was used as a loading control. One of three independent experimental replicates is shown. **(K)** Senescence-associated β-galactosidase (SA-β-Gal) activity, promyelocytic leukemia (PML) body immunofluorescence, and γH2AX and 53BP1 foci. Shown are representative images from one of three independent experimental replicates. Scale bar, 10 μm. **(L)** Quantification of SA-β-Gal activity in K. One of three independent experimental replicates is shown. Data represent mean ± SD ($n = 3$, where each dot represents >100 cells counted). *, P < 0.05 by Student's $t$ test. **(M)** Quantification of PML body immunofluorescence in K. One of three independent experimental replicates is shown. Data represent mean ± SD ($n = 3$, where each dot represents >200 cells counted). *, P < 0.01 by Student's $t$ test. **(N)** Quantification of 53BP1 and γH2AX foci in K. One of three independent experimental replicates is shown. Data represent mean ± SD ($n = 3$, where each dot represents >200 cells counted). *, P < 0.005 by Student's $t$ test. **(O)** C/EBPβ protein expression was determined by immunoblot. Vinculin was used as a loading control. One of six independent experimental replicates is shown. Cont, control; MFI, median fluorescence intensity.

---

that the increase in SASP genes is not an indirect effect of senescence induction by DOT1L. Together with data that DOT1L knockdown does not bypass senescence (Fig. 2), these data demonstrate that DOT1L specifically regulates the SASP and is uncoupled from other senescence phenotypes. cGAS-STING is activated by cytosolic DNA during senescence due to increased DNA damage accumulation and decreased LMNB1 expression (Dou et al., 2017; Glück et al., 2017). Interestingly, we found that overexpression of DOT1L alone increased H3K79me2/3 occupancy at *IL1A* and increased SASP gene expression without inducing DNA damage or altering *LMNB1* expression (Fig. 4, K and N; and Fig. S3 H). Finally, overexpression of DOT1L did not affect p65 phosphorylation, while C/EBPβ was increased at both the RNA and the protein level (Fig. 4 O; and Fig. S3, I and J). Indeed, because C/EBPβ cooperates with the NF-κB transcription complex to allow for the transcriptional activation of downstream SASP genes, such as *CXCL8* (Kuilman et al., 2008; Orjalo et al., 2009), one possibility whereby the SASP expression is significantly lower in DOT1L-overexpressing cells compared with OIS cells is that p65 is not phosphorylated. An additional question remains: How is DOT1L specifically recruited to the *IL1A* gene locus? DOT1L binding to chromatin is promoted by mono-ubiquitination of histone H2B on lysine 120 (H2BK120Ub; Briggs et al., 2002; Kleer et al., 2002; McGinty et al., 2008), and it is possible that H2BK120Ub is elevated at the *IL1A* locus. Indeed, H3K4 methylation, which is also activated by H2BK120Ub (Kim et al., 2009), is upregulated by mixed-lineage leukemia protein-1 at SASP gene loci (Capell et al., 2016). Additionally, DOT1L has multiple binding partners or inhibitors that could affect its recruitment to particular loci, including C/EBPβ itself (Altaf et al., 2007; Cho et al., 2015; Fingerman et al., 2007; Liu et al., 2018; Wakeman et al., 2012; Wood et al., 2018). Future experiments will be aimed at determining the exact mechanism of DOT1L recruitment to the *IL1A* locus in senescence. Nevertheless, these data indicate that DOT1L overexpression is sufficient to modestly induce SASP expression at least in part through increased H3K79me2/3 at the *IL1A* locus and correlates with C/EBPβ expression.

### DOT1L is increased downstream of STING
Finally, we aimed to determine the upstream mediators of DOT1L transcriptional upregulation. Previous studies in yeast have indicated that DOT1L is cell cycle regulated (Feng et al., 2002; Kim et al., 2014); however, this is likely not the case in our model as cells are senescent (Fig. 1, A–E). Reports have demonstrated a role for cGAS in regulating the paracrine effects of the SASP via STING (Dou et al., 2017; Glück et al., 2017; Yang

et al., 2017). A previous report linked the cGAS-STING pathway with toll-like receptor 2 (TLR2; Hari et al., 2019), and DOT1L is upregulated by activation of multiple TLRs (Chen et al., 2020). Therefore, we aimed to determine whether STING is upstream of DOT1L and H3K79 methylation during OIS. STING knockdown decreased *DOT1L* expression and H3K79me2/3 expression in OIS cells (Fig. 5, A–D). As expected, STING knockdown decreased *IL1A* mRNA expression and transcription of downstream cytokines *IL6*, *IL1B*, and *CXCL8* (Fig. 5 E). This was not due to rescue of senescence or changes in DNA damage (Fig. 5, F–L). Rescue experiments using overexpression of DOT1L in STING knockdown OIS cells abrogated the decrease in H3K79me2/3 and *IL1A* expression and rescued the expression of downstream cytokines (Fig. 5, A–E). Together with our previous results, our data implicate the STING pathway in regulating the senescence microenvironment in part through DOT1L-mediated epigenetic control of *IL1A* and further suggest that DOT1L expression participates in a feed-forward mechanism to amplify SASP gene transcription at least in part via C/EBPβ (Fig. 5 M), although the exact mechanism remains to be determined. For instance, future studies are needed to determine whether NF-κB or IRF3, which are both downstream of STING (Chen et al., 2016), directly or indirectly increase *DOT1L*.

In conclusion, our study provides a new understanding of epigenetic regulation of the SASP via the H3K79 methyltransferase DOT1L. This layer of control likely increases expression of SASP genes in a feed-forward pathway that is uncoupled from the senescence-associated cell cycle arrest. As the SASP has been implicated in contributing to detrimental effects of senescence, such as chemoresistance, tumor progression, and age-related pathologies, preventing SASP expression may be therapeutic in a wide range of diseases. Indeed, selective targeting of senescent cells using genetic models or senolytics has clearly demonstrated that inhibition of the SASP in cancer, aging, and other models is beneficial (Bussian et al., 2018; Chandra et al., 2020; Chang et al., 2016; Demaria et al., 2017; Jeon et al., 2017; Justice et al., 2019; Perrott et al., 2017; Suvakov et al., 2019; Wiley et al., 2018; Xu et al., 2015). Our findings provide a rationale for targeting the H3K79 methyltransferase DOT1L to alleviate the harmful effects of the SASP while maintaining a senescent state.

## Materials and methods
### Cells and culture conditions
Normal diploid IMR90 human fibroblasts (CCL-186; ATCC) and BJ-hTERT immortalized human foreskin fibroblasts (a gift from

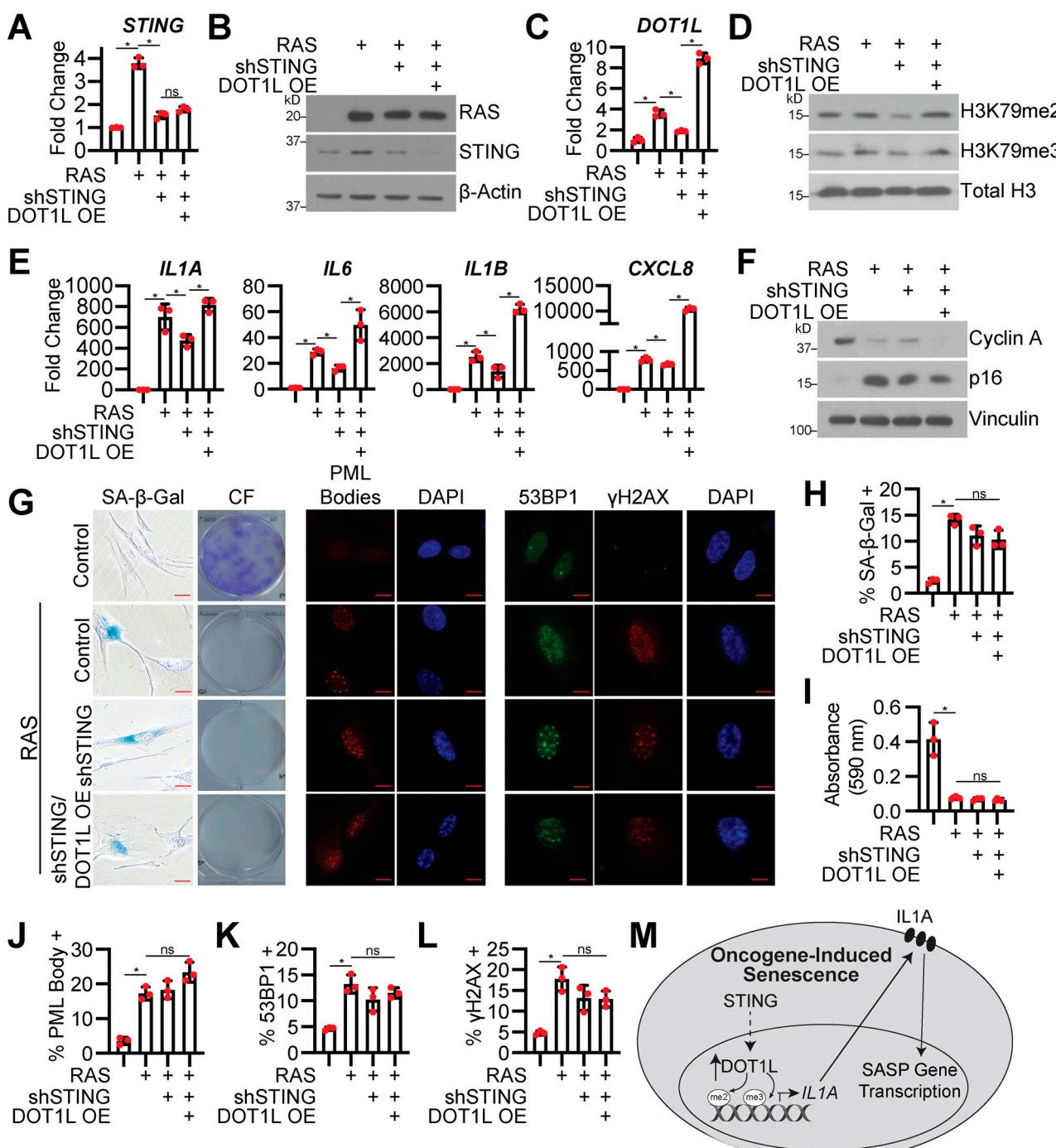

**Figure 5. STING is necessary for DOT1L expression to promote the SASP.** IMR90 cells were infected with retrovirus-expressing HRas^G12V (RAS) or empty vector control with or without lentivirus expressing an shRNA to human STING (shSTING) or shGFP control with or without overexpression of DOT1L (DOT1L OE) or empty vector control. **(A)** *STING* mRNA expression was determined by RT-qPCR. One of three independent experimental replicates is shown. Data represent mean ± SD ($n = 3$). *, $P < 0.0001$ by one-way ANOVA with Tukey's multiple comparisons. **(B)** RAS and STING immunoblot analysis. β-Actin was used as a loading control. One of three independent experimental replicates is shown. **(C)** *DOT1L* mRNA expression was determined by RT-qPCR. One of three independent experimental replicates is shown. Data represent mean ± SD ($n = 3$). *, $P < 0.0001$ by one-way ANOVA with Tukey's multiple comparisons. **(D)** H3K79me2 and H3K79me3 immunoblot analysis on the chromatin fraction. Total histone H3 was used as loading control. One of three independent experimental replicates is shown. **(E)** *IL1A, IL6, IL1B,* and *CXCL8* mRNA expression was determined by RT-qPCR. One of five independent experimental replicates is shown. Data represent mean ± SD ($n = 3$). *, $P < 0.001$ by one-way ANOVA with Tukey's multiple comparisons. **(F)** Immunoblot analysis for the indicated proteins. Vinculin was used as a loading control. One of three independent experimental replicates is shown. **(G)** Senescence-associated β-galactosidase (SA-β-Gal) activity, colony formation (CF), promyelocytic leukemia (PML) body immunofluorescence, and γH2AX and 53BP1 foci. Shown are representative images from one of three independent experimental replicates. Scale bar, 10 μm. **(H)** Quantification of SA-β-Gal in G. One of three independent experimental replicates is shown. Data represent mean ± SD ($n = 3$, where each dot represents >100 cells counted). *, $P < 0.01$ by one-way ANOVA with Tukey's multiple comparisons. **(I)** Quantification of CF in G. One of three independent experimental replicates is shown. Data represent mean ± SD ($n = 3$). *, $P < 0.01$ by one-way ANOVA with

Tukey's multiple comparisons. **(J)** Quantification of PML body foci in G. One of three independent experimental replicates is shown. Data represent mean ± SD ($n$ = 3, where each dot represents >200 cells counted). *, $P < 0.02$ by one-way ANOVA with Tukey's multiple comparisons. **(K)** Quantification of 53BP1 foci in G. One of three independent experimental replicates is shown. Data represent mean ± SD ($n$ = 3, where each dot represents >200 cells counted). *, $P < 0.01$ by one-way ANOVA with Tukey's multiple comparisons. **(L)** Quantification of γH2AX foci in G. One of three independent experimental replicates is shown. Data represent mean ± SD ($n$ = 3, where each dot represents >200 cells counted). *, $P < 0.01$ by one-way ANOVA with Tukey's multiple comparisons. **(M)** Proposed model of DOT1L-mediated SASP induction. Upon OIS, STING induces DOT1L expression, which methylates H3K79 at the *IL1A* locus. IL1A is then transported to the cell surface, where it contributes to the feed-forward mechanism of SASP induction.

Dr. Patricia Opresko, University of Pittsburgh, Pittsburgh, PA) were cultured in 2% $O_2$ in DMEM (10017CV; Corning) with 10% FBS supplemented with L-glutamine, nonessential amino acids, sodium pyruvate, and sodium bicarbonate. IMR90s were used between population-doubling numbers 25–35. HEK293FT (R70007; Thermo Fisher Scientific) and Phoenix cells (a gift from Dr. Gary Nolan, Stanford University, Stanford, CA) were cultured in DMEM (10013CV; Corning) with 5% FBS. Melanocytes (104-05A; Sigma-Aldrich) were cultured in melanocyte growth medium (135-500; Cell Applications). All cell lines were cultured with MycoZap (VZA2032; Lonza) and penicillin-streptomycin (30002Cl; Corning). All cell lines were routinely tested for mycoplasma as described in Uphoff and Drexler (2005). Briefly, cells were cultured for at least 48 h in antibiotic-free media. After 48 h, cell media were collected and pelleted by centrifugation at 1,500 x$g$ for 5 min. Supernatant was transferred to a new tube and boiled for 5 min at 95°C. PCR was performed using a Bio-Rad T100 Thermal Cycler using the following reaction: 0.5 µl of 10 µM forward primer mix (forward primer #1, CGCCTGAGTAGTACGTWCGC; forward primer #2, TGCCTGRGTAGTACATTCGC; forward primer #3, CRCCTGAG-TAGTATGCTCGC; forward primer #4, CGCCTGGGTAGTACATTC GC [R = mixture of G and A; W = mixture of T and A]), 0.5 µl of 10 µM reverse primer mix (reverse primer #1, GCGGTGTGTA-CAARACCCGA; reverse primer #2, GCGGTGTGTACAAACCCCGA [R = mixture of G and A]), 10 µl of Taq 2X Master Mix (M0270L; New England Biolabs), 7 µl of water, and 2 µl of sample media. The following amplification conditions were used: 15 min at 95°C; 30 cycles of 95°C for 30 s, 40 s at 62°C, followed by 30 s at 68°C; and a final extension of 5 min at 70°C. PCR products were analyzed on a 1.5% agarose gel and compared with positive control samples.

### Plasmids and antibodies
pBABE-puro-H-RAS^G12V (39526; Addgene), pBABE-BRAF^V600E (15269; Addgene), MSCB-hDot1Lwt (74173; Addgene), and pHIV-BRAFV600E-mOrange2 (110732; Addgene) were used. pLX304-IL1A (HsCD00441032) was obtained from DNASU Plasmid Repository. pLKO.1-shDOT1L and pLKO.1-shSTING plasmids were obtained from Sigma-Aldrich with the following The RNAi Consortium numbers (TCRNs): shDOT1L #1, TRCN0000020210 (CCGGCCGCAAGAAGAAGCTAAACAACTCGAGTTGTTTAGCTT CTTCTTGCGGTTTTT); shDOT1L #2, TRCN0000020209 (CCG GCGCCAACACGAGTGTTATATTCTCGAGAATATAACACTCGT GTTGGCGTTTTT); and shSTING, TRCN00000160281 (CCGGCA TGGTCATATTACATCGGATCTCGAGATCCGATGTAATATGAC CATGTTTTTTG).

The following antibodies were obtained from the following suppliers: rabbit anti-histone H3 trimethyl K79 (ab208189;

Abcam), rabbit anti-histone H3 dimethyl K79 (ab177184; Abcam), mouse anti-histone H3 (14269; Cell Signaling Technology), rabbit anti-DOT1L (77087; Cell Signaling Technology), rabbit anti-STING (13647; Cell Signaling Technology), rabbit anti-NF-κB-p65 (8242S; Cell Signaling Technology), rabbit anti-phospho-NF-κB-p65 (3033; Cell Signaling Technology), rabbit anti-cyclin A2 (ab181591; Abcam), rabbit anti-CDKN2A/p16INK4a (ab108349; Abcam), rabbit anti-p53 (OP43; Calbiochem), rabbit anti-RAS (610001; BD Transduction Laboratories), rabbit anti-BRAF (sc-5284; Santa Cruz Biotechnology), mouse anti-C/EBPβ (sc-7962; Santa Cruz Biotechnology), mouse anti-V5 Tag (R960-25; Thermo Fisher Scientific), mouse anti-vinculin (V9131; Sigma-Aldrich), mouse anti-β-actin (A1978; Sigma-Aldrich), mouse anti-PML (sc-966; Santa Cruz Biotechnology), mouse anti-γH2AX (05-636; EMD Millipore), rabbit anti-53BP1 (A300-272A; Bethyl Laboratories), mouse anti-IgG1 kappa (50-186-16; Thermo Fisher Scientific), and mouse anti-IL-1α (11-7118-81; Thermo Fisher Scientific). For Western blotting, secondary antibodies used were HRP-linked anti-mouse IgG (7076S, 1:8,000 dilution; Cell Signaling Technology) or HRP-linked anti-rabbit IgG (7074S, 1:5,000 dilution; Cell Signaling Technology). For immunofluorescence, secondary antibodies used were Cy3 AffiniPure Donkey anti-mouse IgG (715-165-150, 1:5,000 dilution; Jackson ImmunoResearch) or fluorescein AffiniPure Donkey anti-Rabbit IgG (711-095-152, 1:2,000 dilution; Jackson ImmunoResearch).

### Retroviral and lentiviral transduction and infection
Phoenix cells (a gift from Gary Nolan, Stanford University, Stanford, CA) were used to package retroviral infection viruses. Retrovirus production and transduction were performed using the BES-buffered saline (BBS)/$CaCl_2$ method as described in Aird et al. (2013). Briefly, 30 µg of plasmid DNA was added to 0.5 mL of 0.25 M $CaCl_2$, and 0.5 mL of 2× BBS (50 mM N,N-bis(2-hydroxyethyl)-2-aminoethanesulfonic acid, 280 mM NaCl, 1.5 mM $Na_2HPO_4$) was added in dropwise manner to the DNA/$CaCl_2$ mixture and incubated at room temperature for 15 min. The DNA/$CaCl_2$/BBS mixture was added dropwise to QNX in 9 ml fresh media cells. Media were changed after 24 h, and first and second round viral particles were collected and filtered 48 h and 72 h after transfection, respectively. Cells were infected with two rounds of 3 ml viral supernatant, 3 ml fresh media, and 6 µl of 8 mg/ml polybrene (H9268; Sigma-Aldrich). HEK293FT cells were used to package lentiviral viruses using the ViraPower kit (Invitrogen). Briefly, 3 µl of each pLP1, pLP2, and pLPV-SVG were placed in 3 ml of Opti-MEM Reduced Serum Medium (BRL 31985070; Gibco), and 36 µg of polyethylenimine (239661; Polysciences) was added to the plasmid and media mixture and incubated at room temperature for 20 min. Gently, 3 ml of

transfection mix was added to cells for 6 h. After 6 h, media on cells were replaced with DMEM (10013CV; Corning) supplemented with 5% FBS. Viral supernatant was collected 48 h later. For lentivirus infections, 2 ml of viral supernatant, 4 ml of cell media, and 6 µl of 8 mg/ml polybrene was used. IMR90 and BJ-hTERT cells were infected with retrovirus and/or lentivirus as indicated in Fig. S1 A. IMR90 and BJ-hTERT cells were infected with two rounds of pBABE-Control, pBABE-HRAS^G12V, or pBABE-BRAF^V600E. Cells were selected with 1 µg/ml puromycin for 7 d. IMR90 and BJ-hTERT cells were also infected with pLKO.1-shGFP (control), pLKO.1-shDOT1L, or pLKO.1-shSTING where indicated and selected with 3 µg/ml puromycin for 4 d. Alternatively, IMR90 and BJ-hTERT cells were infected with two rounds of MSCB-hDot1lwt and selected with 1 µg/ml blasticidin for 4 d. IMR90 cells were infected with pBABE-Control or pBABE-HRAS^G12V following simultaneous infection with pLKO.1-shGFP (control) or pLKO.1-shDOT1L and pLX304-null (control) or pLX304-IL1A. Cells were selected with 3 µg/ml puromycin for 4 d. For the rescue experiment (Fig. 5), IMR90 cells were initially infected with pBABE-Control or pBABE-HRAS^G12V followed by a simultaneous infection with MSCB-hDot1lwt and pLKO.1-shGFP or pLKO.1-shSTING. Cells were selected with puromycin and blasticidin for 7 d. Melanocyte cells were infected with pLKO.1-shGFP or pHIV-BRAF^V600E-mOrange2 followed by collection for experiments 30 d after infection.

### Mass spectrometry analysis of histone modifications

The cell pellet was resuspended in nuclear isolation buffer (15 mM Tris-HCl, pH 7.5, 60 mM KCl, 15 mM NaCl, 5 mM MgCl$_2$, 1 mM CaCl$_2$, 250 mM sucrose, 1 mM DTT, 1:100 Halt Protease Inhibitor Cocktail [78430; Thermo Scientific], and 10 mM sodium butyrate). Nuclei were resuspended in 0.2 M H$_2$SO$_4$ for 1 h at room temperature and centrifuged at 4,000 ×$g$ for 5 min. Histones were precipitated from the supernatant by the addition of TCA at a final concentration of 20% TCA (vol/vol). Precipitated histones were pelleted at 10,000 ×$g$ for 5 min, washed once with 0.1% HCl in acetone and twice with acetone, followed by centrifugation at 14,000 ×$g$ for 5 min. Histones were air dried then resuspended in 10 µl of 0.1 M (NH)$_4$HCO$_3$ for derivatization and digestion according to Garcia et al. (2007). Peptides were resuspended in 100 µl 0.1% trifluoroacetic acid in water for liquid chromatography–tandem mass spectrometry (LC-MS/MS) analysis.

Multiple reaction monitoring was performed on a triple quadrupole mass spectrometer (TSQ Quantiva; Thermo Fisher Scientific) coupled with an UltiMate 3000 Dionex nano-LC system. Peptides were loaded with 0.1% trifluoroacetic acid in water at 2.5 µl/min for 10 min onto a trapping column (3 cm × 150 µm, Bischoff ProntoSIL C18-AQ, 3 µm, 200 Å resin) and then separated on a New Objective PicoChip analytical column (10 cm × 75 µm, ProntoSIL C18-AQ, 3 µm, 200 Å resin). Separation of peptides was achieved using solvent A (0.1% formic acid in water) and solvent B (0.1% formic acid in 95% acetonitrile) with the following gradient: 0%–35% solvent B at a flow rate of 0.30 µl/min over 45 min. The following triple quadrupole settings were used across all analyses: collision gas pressure of 1.5 mTorr, Q1 peak width of 0.7 (full width half maximum), cycle time of 2 s, skimmer

offset of 10 V, and electrospray voltage of 2.5 kV. Monitored peptides were selected based on previous reports (Zheng et al., 2012; Zheng et al., 2013).

Raw MS files were imported and analyzed in Skyline software with Savitzky-Golay smoothing (MacLean et al., 2010). Automatic peak assignments from Skyline were manually confirmed. Peptide peak areas from Skyline were used to determine the relative abundance of each histone modification by calculating the peptide peak area for a peptide of interest and dividing by the sum of the peak areas for all peptides with that sequence. The relative abundances were determined based on the mean of three technical replicates, with error bars representing the standard deviation.

### Western blotting

Cell lysates were collected in sample buffer (2% SDS, 10% glycerol, 0.01% bromophenol blue, 62.5 mM Tris, pH 6.8, 0.1 M DTT), boiled to 95°C for 10 min, and sonicated. Chromatin samples were collected by trypsinizing and pelleting cells in 1× PBS at 3,000 rpm for 4 min. Cell pellets were washed with 1× PBS and pelleted again at 3,000 rpm for 4 min. Cell pellets were resuspended in a 10 mM Hepes-KOH (pH 8.0), 10 mM KCl, 1.5 mM MgCL$_2$, 0.34 M sucrose, 10% glycerol, 1 mM DTT, 0.1 mM PMSF, and proteinase inhibitor cocktail buffer with a final pH of 7.5 (buffer A). 0.1% Triton X-100 was added to each sample and incubated on ice for 5 min. Samples were then centrifuged at 1,300 ×$g$ for 4 min at 4°C. Following centrifugation, buffer was removed followed by addition of buffer A and centrifugation at 1,300 ×$g$ for 4 min at 4°C two additional times. Buffer A was completely removed, and cell pellets were resuspended in 3 mM EDTA (pH 8.0), 0.2 mM EGTA, 1 mM DTT, 0.1 mM PMSF, and proteinase inhibitor cocktail buffer with a final pH of 8.0 (buffer B) and incubated on ice for 30 min. Samples were then centrifuged at 1,700 ×$g$ for 4 min at 4°C. Following centrifugation, buffer was removed followed by addition of buffer B and centrifugation at 1,700 ×$g$ for 4 min at 4°C two additional times. Buffer B was aspirated, and a half volume of 3× sample buffer and 1 volume of 1× sample buffer were added to each chromatin pellet.

Total protein and chromatin concentrations were obtained using the Bradford assay. An equal amount of total protein or chromatin was resolved using SDS-PAGE gels and transferred onto nitrocellulose membranes. Membranes were blocked with 5% nonfat milk or 4% BSA in TBS containing 0.1% Tween 20 (TBS-T) for 1 h at room temperature. Membranes were incubated overnight at 4°C in primary antibodies in 4% BSA/TBS and 0.025% sodium azide. Membranes were then washed four times in TBS-T for 5 min at room temperature followed by an incubation with the HRP-conjugated secondary antibodies listed above. Membranes were washed four times in TBS-T for 5 min at room temperature and visualized on film with SuperSignal West Pico PLUS Chemiluminescent Substrate (34580; Thermo Fisher Scientific).

### ChIP and ChIP-seq

ChIP was performed as previously described in (Aird et al., 2016) using the ChIP-grade antibodies described above. Cells were

fixed in 1% paraformaldehyde for 5 min at room temperature and quenched with 1 mL of 2.5 M glycine for 5 min at room temperature. Cells were washed twice with cold PBS. Cells were lysed in 1 ml ChIP lysis buffer (50 mM Hepes-KOH, pH 7.5, 140 mM NaCl, 1 mM EDTA, pH 8.0, 1% Triton X-100, and 0.1% deoxycholate [DOC] with 0.1 mM PMSF and EDTA-free Protease Inhibitor Cocktail [P8340; Sigma-Aldrich]). Samples were incubated on ice for 10 min and then centrifuged at 3,000 rpm for 3 min at 4°C. The pellet was resuspended in 500 µl lysis buffer 2 (10 mM Tris, pH 8.0, 200 mM NaCl, 1 mM EDTA, and 0.5 mM EGTA with 0.1 mM PMSF and EDTA-free Protease Inhibitor Cocktail) and incubated at room temperature for 10 min. Samples were centrifuged at 3,000 rpm for 5 min at 4°C. Next, the pellet was resuspended in 300 µl lysis buffer 3 (10 mM Tris, pH 8.0, 100 mM NaCl, 1 mM EDTA, 0.5 mM EGTA, 0.1% DOC, and 0.5% N-lauroylsarcosine with 0.1 mM PMSF and EDTA-free Protease Inhibitor Cocktail). Cells were sonicated while on ice (10 s on, 50 s off). Next, 30 µl of 10% Triton X-100 was added to each tube, and samples were centrifuged at maximum speed for 15 min at 4°C. The supernatant was transferred to new tubes, and the DNA concentration was quantified. Samples were pre-cleared for 1 h at 4°C on a rotator using 15 µl protein G Dynabeads (Thermo Fisher Scientific) in ChIP lysis buffer. Samples were centrifuged at maximum speed for 15 min at 4°C, after which the supernatant was transferred to a new tube. 50 µl of the antibody bead conjugate solution was added, and chromatin was immunoprecipitated overnight on a rotator at 4°C. The following washes were performed twice for 15 min each by rotating for 15 min at 4°C: ChIP lysis buffer, ChIP lysis buffer plus 0.65 M NaCl, wash buffer (10 mM Tris-HCl, pH 8.0, 250 mM LiCl, 0.5% NP-30, 0.5% DOC, and 1 mM EDTA, pH 8.0), and 10 mM Tris-HCl (pH 8.0) and 1 mM EDTA (pH 8.0). DNA was eluted by incubating the beads with fresh 50 mM Tris-HCl (pH 8.0), 10 mM EDTA (pH 8.0), and 1% SDS for 30 min at 65°C. Reversal of cross-linking was performed by incubating samples overnight at 65°C. Proteins were digested using 1 mg/ml proteinase K and incubating at 37°C for 5 h. Finally, the DNA was purified using the Wizard SV Gel and PCR Clean Up kit (A9282; Promega).

Immunoprecipitated DNA was analyzed by qPCR using iTaq Universal SYBR® Green Supermix (1725121; Bio-Rad). The following amplification conditions were used: 5 min at 95°C and 40 cycles of 95°C for 10 s and 30 s with 62°C annealing temperature. The assay ended with the following melting curve program: 15 s at 95°C, 1 min at 60°C, then ramping to 95°C while continuously monitoring fluorescence. All samples were assessed in triplicate. Enrichment of H3K79me2, H3K79me3, and DOT1L was determined and normalized to total histone H3. ChIP-qPCR primer sets are described in Table S4.

IMR90 cells induced to senescence by oncogenic RAS were prepared and analyzed through ChIP-seq using anti-histone H3 trimethyl K79 (ab208189; Abcam). ChIP-seq libraries were created using Kapa HyperPrep Kit (Roche Sequencing and Life Science). The unique dual index sequences (NEXTFLEX Unique Dual Index Barcodes, Bioo Scientific) were incorporated in the adaptors for multiplexed high-throughput sequencing. The final product was assessed for its size distribution and concentration

using BioAnalyzer High Sensitivity DNA Kit (Agilent Technologies). The libraries were pooled and diluted to 3 nM using 10 mM Tris-HCl (pH 8.5) and then denatured using the Illumina protocol. The denatured libraries were loaded onto an S1 flow cell on an Illumina NovaSeq 6000 and run for 2 × 50 cycles according to the manufacturer's instructions. Demultiplexed and adapter-trimmed sequencing reads were generated using Illumina bcl2fastq (released version 2.20.0.422). BBDuk (https://sourceforge.net/projects/bbmap) was used to trim/filter low-quality sequences using the "qtrim=lr trimq=10 maq=10" option. Next, alignment of the filtered reads to the human reference genome (GRCh38) was done using Bowtie 2 (version 2.3.4.3; Langmead and Salzberg, 2012). Only sequences that aligned uniquely to the human genome were used to identify peaks. Peaks were called using MACS2 (version 2.1.1) and annotated using HOMER (version 4.10) by supplying GRCh38.78.gtf as the annotation file.

For the ChIP-seq analysis, raw ChIP-seq data reads were aligned against the hg19 version of human genome using Bowtie (Langmead et al., 2009), and HOMER (Heinz et al., 2010) was used to generate BIGWIG files and call significant peaks versus input using the "-style histone" option. Peaks that passed the fourfold, FDR <5% threshold were considered. Normalized signals for significant peaks were derived from BIGWIG files using the bigWigAverageOverBed tool from the University of California, Santa Cruz, toolbox (Kent et al., 2010) with mean0 option. Fold differences between samples were then calculated with average input signal 0.4 used as a floor for the minimum allowed signal. Genes with transcript body overlapping significant peaks were considered to be putatively regulated by the histone modification. Processed data can be found in Table S5. Raw peak counts and analysis generated during this study are available from GEO accession no. GSE156591.

## RNA-seq

A total of 1 µg of RNA was used for cDNA library construction at Novogene using an NEBNext Ultra 2 RNA Library Prep Kit for Illumina (E7775; New England Biolabs). Briefly, mRNA was enriched using oligo(dT) beads followed by two rounds of purification and fragmented randomly by adding fragmentation buffer. The first-strand cDNA was synthesized using random hexamers primer, after which a custom second-strand synthesis buffer (Illumina), dNTPs, RNase H, and DNA polymerase I were added to generate the second strand (double-stranded cDNA). After a series of terminal repair, polyadenylation, and sequencing adaptor ligation, the double-stranded cDNA library was completed after size selection and PCR enrichment. The resulting 250–350-bp insert libraries were quantified using a Qubit 2.0 fluorometer (Thermo Fisher Scientific) and qPCR. Size distribution was analyzed using an Agilent 2100 Bioanalyzer. Qualified libraries were sequenced on an Illumina NovaSeq 6000 Platform using a paired-end 150 run (2 × 150 bases). 20 million raw reads were generated from each library.

For RNA-seq analysis, reference genome and gene model annotation files were downloaded from genome website browser (National Center for Biotechnology Information/University of California, Santa Cruz/Ensembl) directly. Indexes of the reference

genome were built using STAR, and paired-end clean reads were aligned to the reference genome using STAR (version 2.5). STAR uses the method of maximal mappable prefix, which can generate a precise mapping result for junction reads. STAR counts the number of reads per gene while mapping, which coincide with those produced by htseq-count with default parameters. The fragments per kilobase of transcript per million (FPKM) of each gene was calculated based on the length of the gene and reads count mapped to this gene. Differential expression analysis between two conditions/groups (two biological replicates per condition) was performed using the DESeq2 R package (version 1.14.1). The resulting P values were adjusted using Benjamini and Hochberg's approach for controlling the FDR. Genes with an adjusted P value (FDR) < 0.05 found by DESeq2 were assigned as differentially expressed. Processed data can be found in Table S6. Raw counts and differential expression analysis generated during this study are available at GEO accession no. GSE156648.

### Gene set enrichment analysis

Common genes between ChIP-seq and RNA-seq (FC >1.5 RAS versus control) were analyzed using the Broad Institute Gene Set Enrichment Analysis (GSEA) webtool (https://www.gsea-msigdb.org/gsea/msigdb/annotate.jsp) for Hallmarks and Reactome (Liberzon et al., 2015; Subramanian et al., 2005). Following GSEA documentation, indication terms were considered significant when the FDR-adjusted P value (q-value) was <0.25.

### RT-qPCR

Total RNA was extracted from cells using TRIzol and DNase treated, cleaned, and concentrated using Zymo columns (R1013; Zymo Research). Optimal density values of extracted RNA were obtained using NanoDrop One (Thermo Fisher Scientific). Only RNA with A260 and A280 ratios >1.9 was used. Relative expression of target genes listed in Table S4 were analyzed using the QuantStudio 3 Real-Time PCR System (Thermo Fisher Scientific). All primers were designed using the Integrated DNA Technologies tool (http://eu.idtdna.com/scitools/Applications/RealTimePCR; Table S4). A total of 25 ng RNA was used in One-Step qPCR (95089; Quanta BioSciences). The following amplification conditions were used: 10 min at 48°C, 5 min at 95°C, and 40 cycles of 10 s at 95°C and 7 s at the annealing temperature of 62°C. The assay ended with a melting curve program as follows: 15 s at 95°C, 1 min at 70°C, then ramping to 95°C while continuously monitoring fluorescence. All samples were assessed in triplicate. Relative quantification was determined while normalizing to multiple reference genes (*MRPL9*, *PUM1*, *PSMC4*) using the delta-delta Ct method.

### Cell surface staining of IL1A

Cells were washed two times with 1× PBS and harvested using trypsin. Cells were pelleted by centrifugation at 1,000 rpm for 5 min at 4°C and then washed twice with 3% BSA/PBS. Cells were stained (10 μl/1.5 × 10^5 cells) with anti-IL1α (11-7118-81; Thermo Fisher Scientific) and anti-IgG1 kappa (50-186-16; Thermo Fisher Scientific) for 1 h on ice. An unstained control was also incubated on ice for 1 h. Cells were then pelleted by centrifugation at 1,000 rpm for 5 min at 4°C and washed with 3% BSA/PBS. Cell pellets were resuspended in 200 μl 3% BSA/PBS. Stained cells and unstained control cells were run on a 10-color FACSCanto flow cytometer (BD Biosciences). All data were analyzed using FlowJo software.

### Cytokine secretion quantification

The human cytokine antibody array C1000 (AAH-CYT-1000-2; RayBio) was used to quantify secreted factors. Conditioned media were obtained from cells that were washed with 1× PBS and incubated in serum-free media for 48 h. A 0.2-μm filter was used to filter conditioned media. Membranes were visualized on film. Individual spot signal densities were obtained using ImageJ software and normalized to cell number from which the conditioned media were obtained.

### Senescence-associated β-galactosidase activity assay

Cells were seeded at an equal density 24 h before assay. Cells were fixed for 5 min at room temperature using 2% formaldehyde/0.2% glutaraldehyde in 1× PBS. Cells were then washed twice with 1× PBS and stained with 40 mM $Na_2H\ PO_4$, 150 mM NaCl, 2 mM $MgCl_2$, 5 mM $K_3Fe(CN)_6$, 5 mM $K_4Fe(CN)_6$, and 1 mg/ml X-gal staining solution overnight at 37°C in a non-$CO_2$ incubator. At least 100 cells per well were counted. Brightfield images were taken a room temperature using a Nikon Ts2 with a 20×/0.40 objective (Nikon LWD) equipped with a camera (Nikon DS-Fi3) and acquired using NIS-Elements D software.

### Immunofluorescence

Cells were seeded at an equal density on coverslips and fixed with 4% paraformaldehyde. Cells were washed four times with 1× PBS and permeabilized with 0.2% Triton X-100 in PBS for 5 min and then postfixed with 1% paraformaldehyde and 0.01% Tween 20 for 30 min. Cells were blocked for 5 min with 3% BSA/PBS followed by incubation of corresponding primary antibody in 3% BSA/PBS for 1 h at room temperature. Prior to incubation with secondary antibody in 3% BSA/PBS for 1 h at room temperature, cells were washed three times with 1% Triton X-100 in PBS. Cells were then incubated with 0.15 μg/ml DAPI in 1× PBS for 1 min, washed three times with 1× PBS, mounted with fluorescence mounting medium (9 ml of glycerol [BP229-1; Fisher Scientific], 1 ml of 1× PBS, and 10 mg of p-phenylenediamine [PX0730; EMD Chemicals]; pH was adjusted to 8.0–9.0 using carbonate-bicarbonate buffer [0.2 M anhydrous sodium carbonate, 0.2 M sodium bicarbonate]) and sealed. At least 200 cells per coverslip were counted. Images were obtained at room temperature using a Nikon ECLIPSE 90i microscope with a 20×/0.17 objective (Nikon DIC N2 Plan Apo) equipped with a camera (DS-Qi1). Images were acquired using NIS-Elements AR software and processed using ImageJ.

### Colony formation assay

Cells were seeded at an equal density in 6-well plates and cultured for 14 d. Cells were then fixed using 1% paraformaldehyde/PBS and stained with 0.05% crystal violet. 10% acetic acid was used to destain, and absorbance was measured at 590 nm using a SpectraMax 190 spectrophotometer.

## DMBA/TPA mouse treatment

The shaved back skin of female WT FVB/NJ mice between 7 and 8 wk of age was treated topically with a single dose of DMBA in acetone (25 µg in 250 µl). Beginning 1 wk after DMBA exposure, the mice were treated topically with twice-weekly doses of TPA in acetone (5 µg in 250 µl) for 15 wk. The mice were euthanized 16 wk after DMBA treatment. Papillomas were removed, and RNA was isolated by Tri Reagent extraction (TR118; Molecular Research Center).

## Statistical analysis

Data shown are three independent technical replicates performed from one independent experiment. At least three independent biological replicates were performed for each experiment shown unless specified otherwise. All statistical analyses were performed using GraphPad Prism 8.3.1 software. The appropriate statistical test was used as indicated to determine P values of raw data. For one-way ANOVA, post hoc analysis using Tukey's multiple comparisons was performed. $P < 0.05$ was considered significant.

## Online supplemental material

Fig. S1 (complementary to Fig. 1) shows that other senescent cells also display increased H3K79 methylation and DOT1L expression, corresponding to increased *IL1A* and *KDM2B/KDM4D*, are unaltered in RAS-induced senescent cells. Fig. S2 (complementary to Fig. 2) demonstrates that knockdown of DOT1L in BRAF-induced senescent cells is similar to RAS-induced senescence. Fig. S3 (complementary to Figs. 3 and 4) shows that DOT1L overexpression in BJ-hTERT cells increases the SASP but does not affect other senescence markers and that p-p65 expression is not affected by DOT1L overexpression. Table S1 shows the raw data for the epiproteomics analysis in RAS versus control cells. Table S2 lists RAS versus control genes with peaks (FC >1.5) in ChIP-seq and upregulated (FC >1.5; FDR <0.25) in RNA-seq. Table S3 shows the integrated density of cytokines from antibody array. Table S4 lists primers used for the studies. Table S5 shows the annotated ChIP-seq peaks and analysis. Table S6 shows the RNA-seq differential expression analysis.

## Acknowledgments

This work was supported by grants from the National Institutes of Health (F31CA250366 to K.E. Leon, F31CA236372 to E.S. Dahl, and R00CA194309 and R37CA240625 to K.M. Aird), the W. W. Smith Charitable Trust (to K.M. Aird), and the Penn State Cancer Institute postdoctoral fellowship (to R. Buj). Histone epiproteomics services were performed by the Northwestern Proteomics Core Facility, generously supported by National Cancer Institute Cancer Center Support Grant P30 CA060553 awarded to the Robert H. Lurie Comprehensive Cancer Center and NIH Office of the Director instrumentation award S10OD025194, and the National Resource for Translational and Developmental Proteomics supported by National Institutes of Health grant P41 GM108569.

The authors declare no competing financial interests.

Author contributions: Conceptualization: K.E. Leon and K.M. Aird. Investigation: K.E. Leon, R. Buj, E. Lesko, E.S. Dahl, C.-W. Chen, and N. Kumar Tangudu. Formal analysis: K.E. Leon, R. Buj, Y. Imamura, A.V. Kossenkov, and K.M. Aird. Methodology: Y. Imamura and A.V. Kossenkov. Visualization: K.E. Leon, R. Buj, and K.M. Aird. Supervision: R.P. Hobbs and K.M. Aird. Writing–original draft: K.E. Leon and K.M. Aird. Writing–review and editing: K.E. Leon, R. Buj, N. Kumar Tangudu, and K.M. Aird. Funding acquisition: A.V. Kossenkov, R.P. Hobbs, and K.M. Aird.

Submitted: 21 August 2020

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

# Supplemental material

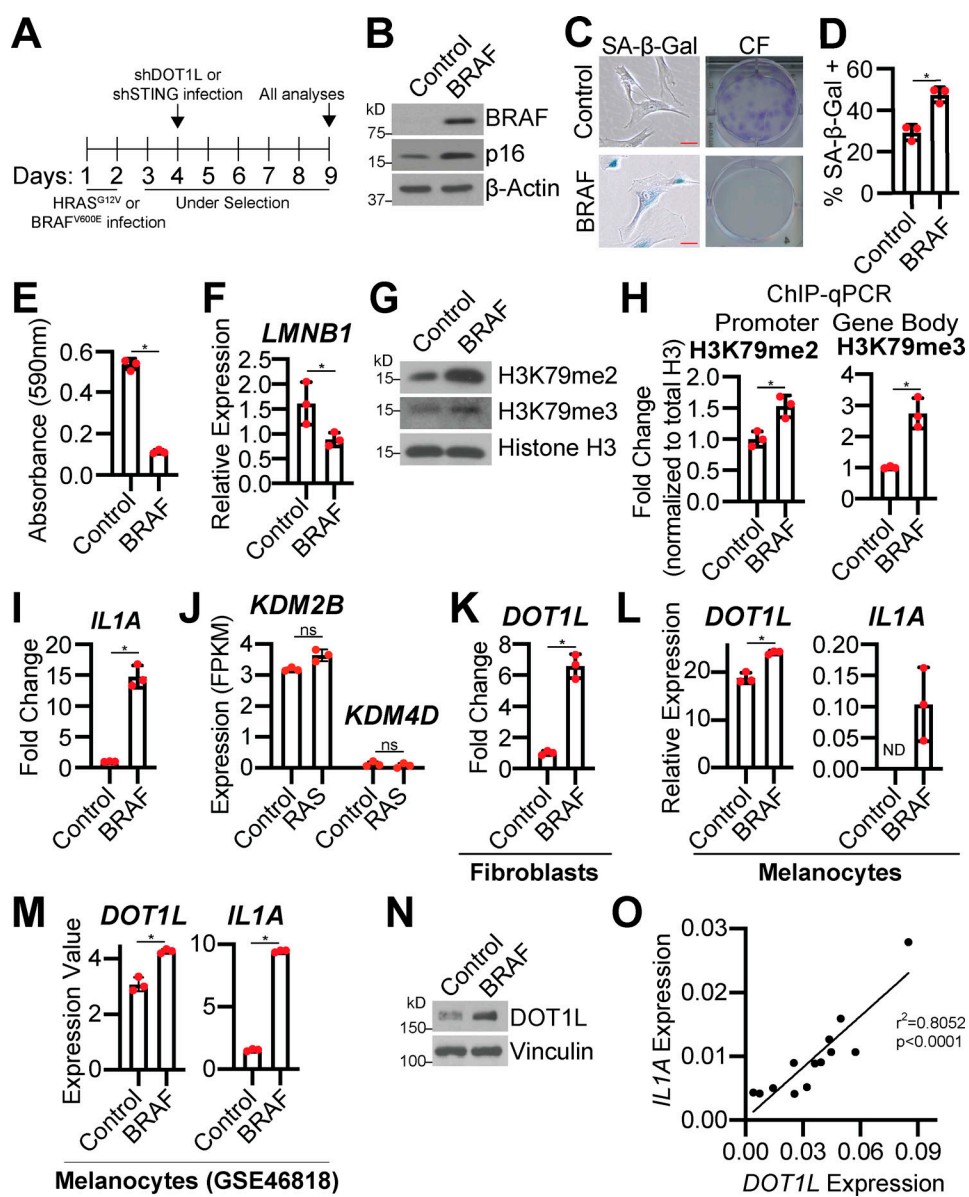

**Figure S1. Oncogenic BRAF induces cellular senescence and increases DOT1L expression; RAS does not affect *KDM2B* or *KDM4D* expression.** Related to Fig. 1. **(A)** Timeline of experiments. **(B–I, K, and N)** BJ-hTERT cells were infected with retrovirus-expressing BRAF^V600E (BRAF) or empty vector control. **(B)** Immunoblot analysis of the indicated proteins. β-Actin was used as a loading control. One of three independent experimental replicates is shown. **(C)** Senescence-associated β-galactosidase (SA-β-Gal) activity and colony formation (CF). Shown are representative images from one of three independent experimental replicates. Scale bar, 10 μm. **(D)** Quantification of SA-β-Gal activity in C. One of three independent experimental replicates is shown. Data represent mean ± SD (*n* = 3, where each dot represents >100 cells counted). *, P < 0.0004 by Student's *t* test. **(E)** Quantification of CF in C. One of three independent experimental replicates is shown. Data represent mean ± SD (*n* = 3). *, P < 0.0008 by Student's *t* test. **(F)** *LMNB1* mRNA expression. One of three independent experimental replicates is shown. Data represent mean ± SD (*n* = 3). *, P < 0.01 by Student's *t* test. **(G)** H3K79me2 and H3K79me3 immunoblot analysis of chromatin fraction. Total histone H3 was used as a loading control. One of three independent experimental replicates is shown. **(H)** H3K79me2 binding to the *IL1A* promoter region and H3K79me3 binding to the gene body was determined by ChIP-qPCR and normalized to total histone H3 binding at the same site. One of two independent experimental replicates is shown. Data represent mean ± SD (*n* = 3). *, P < 0.0140 by Student's *t* test. **(I)** *IL1A* mRNA expression. One of five independent experimental replicates is shown. Data represent mean ± SD (*n* = 3). *, P < 0.0002 by Student's *t* test. **(J)** IMR90 cells were infected with retrovirus-expressing HRas^G12V (RAS) or empty vector control. *KDM2B* and *KDM4D* mRNA expression from RNA-seq. Three technical replicates from one experiment are shown. Data represent mean ± SD (*n* = 3). Not significant by Student's *t* test. **(K)** *DOT1L* mRNA expression. One of three independent experimental replicates is shown. Data represent mean ± SD (*n* = 3). *, P < 0.05 by Student's *t* test. **(L and M)** Primary human melanocytes were infected with retrovirus-expressing BRAF^V600E (BRAF) or control. **(L)** *DOT1L* and *IL1A* mRNA expression was determined by RT-qPCR. One independent experiment is shown. Data represent mean ± SD (*n* = 3). *, P < 0.005 by Student's *t* test. **(M)** *DOT1L* and *IL1A* mRNA expression (GEO accession no. GSE46818). Data represent mean ± SD (*n* = 3). *, P < 0.005 by Student's *t* test. **(N)** DOT1L immunoblot analysis. Vinculin was used as a loading control. One of three independent experimental replicates is shown. **(O)** RT-qPCR analysis was performed for *DOT1L* and *IL1A* expression on papillomas from mice treated with DMBA/TPA. *r²* is the Pearson's correlation coefficient (*n* = 13). ND, not detected.

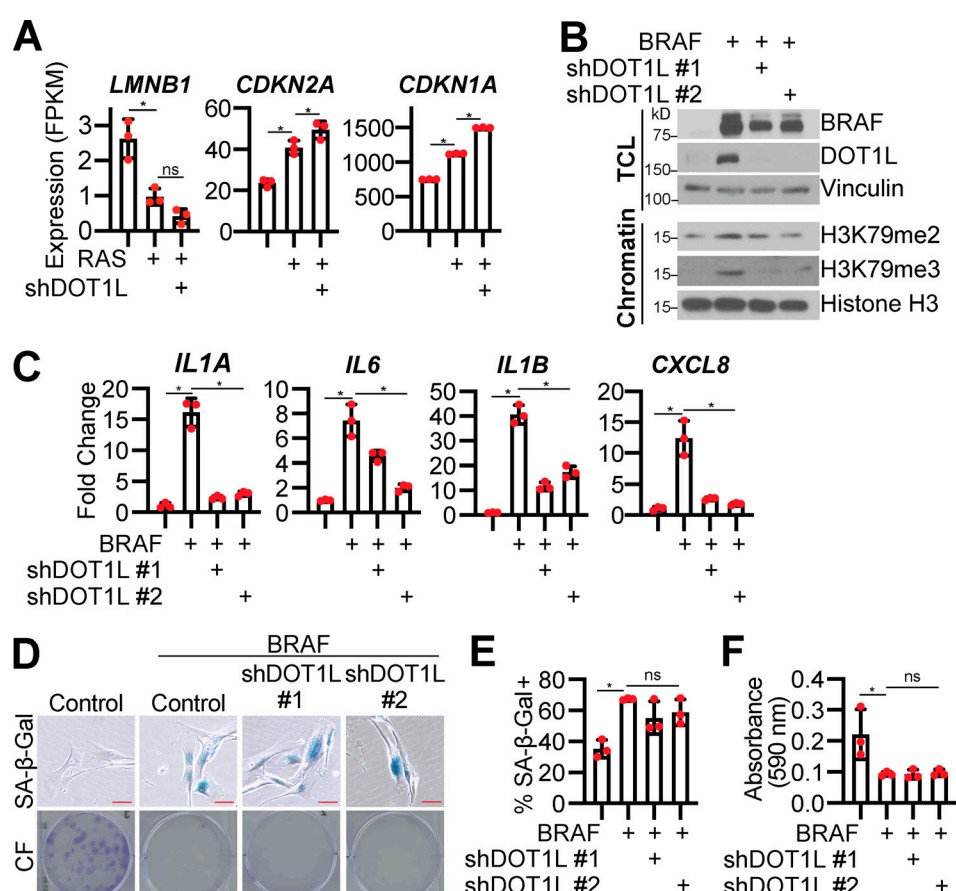

Figure S2. **DOT1L knockdown in BRAF-induced senescent cells decreases SASP and maintains cells in a senescence-associated cell cycle arrest.** Related to Figs. 2 and 3. **(A)** IMR90 cells were infected with retrovirus-expressing HRas^G12V (RAS) or empty vector control. *LMNB1*, *CDKN2A*, and *CDKN1A* mRNA expression from RNA-seq. Three technical replicates from one experiment are shown. Data represent mean ± SD (*n* = 3). *, P < 0.05 by one-way ANOVA with Tukey's multiple comparisons. **(B–F)** BJ-hTERT cells were infected with retrovirus-expressing BRAF^V600E (BRAF) or empty vector control with or without lentivirus-expressing an shRNA to human DOT1L (shDOT1L) or shGFP control. **(B)** Immunoblot analysis of total cell lysates (TCL) and chromatin fractions of the indicated proteins. Vinculin was used as a loading control for TCL. Histone H3 was used as a loading control for chromatin fractions. One of three independent experimental replicates is shown. **(C)** *IL1A*, *IL6*, *IL1B*, and *CXCL8* mRNA expression was determined by RT-qPCR. One of three independent experimental replicates is shown. Data represent mean ± SD (*n* = 3). *, P < 0.05 by one-way ANOVA with Tukey's multiple comparisons. **(D)** Senescence-associated β-galactosidase (SA-β-Gal) activity and colony formation (CF). Shown are representative images from one of three experimental replicates. Scale bar, 10 μm. **(E)** Quantification of SA-β-Gal activity in D. One of three independent experimental replicates is shown. Data represent mean ± SD (*n* = 3, where each dot represents >100 cells counted). *, P < 0.05 by one-way ANOVA with Tukey's multiple comparisons. **(F)** Quantification of CF in D. One of three independent experimental replicates is shown. Data represent mean ± SD (*n* = 3). *, P < 0.05 by one-way ANOVA with Tukey's multiple comparisons.

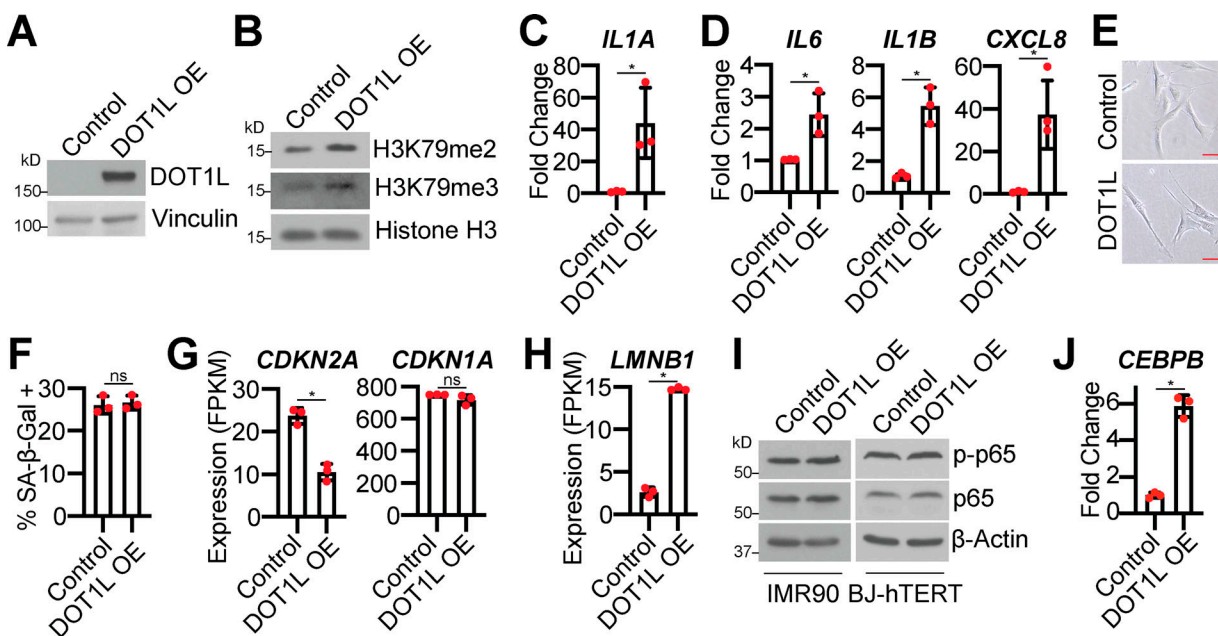

Figure S3. **DOT1L overexpression (OE) in BJ-hTERT cells induces the SASP but not a senescence-associated cell cycle arrest or DNA damage accumulation; p-p65 is not affected by DOT1L OE.** Related to Fig. 4. **(A–F and I)** BJ-hTERT cells were infected with retrovirus-expressing human DOT1L or empty vector control. **(A)** Immunoblot analysis of DOT1L. Vinculin was used as a loading control. One of three independent experimental replicates is shown. **(B)** H3K79me2 and H3K79me3 immunoblot analysis on chromatin fractions. Histone H3 was used as a loading control. One of three independent experimental replicates is shown. **(C)** *IL1A* mRNA expression was determined by RT-qPCR. One of three independent experimental replicates is shown. Data represent mean ± SD ($n$ = 3). *, P < 0.0001 by Student's $t$ test. **(D)** *IL6*, *IL1B*, and *CXCL8* mRNA expression was determined by RT-qPCR. One of three independent experimental replicates is shown. Data represent mean ± SD ($n$ = 3). *, P < 0.0001 by Student's $t$ test. **(E)** Senescence-associated β-galactosidase (SA-β-Gal) activity. Shown are representative images from one of three independent experimental replicates. Scale bar, 10 μm. **(F)** Quantification of SA-β-Gal in E. One of three independent experimental replicates is shown. Data represent mean ± SD ($n$ = 3, where each dot represents >100 cells counted). Not significant by Student's $t$ test. **(G–J)** IMR90 cells were infected with retrovirus-expressing human DOT1L or empty vector control. **(G)** *CDKN2A* and *CDKN1A* mRNA expression from RNA-seq. Three technical replicates from one experiment are shown. Data represent mean ± SD ($n$ = 3). *, P < 0.005 by Student's $t$ test. **(H)** *LMNB1* expression from RNA-seq. Three technical replicates from one experiment are shown. Data represent mean ± SD ($n$ = 3). *, P < 0.0001 by Student's $t$ test. **(I)** Immunoblot analysis of p-p65 and total p65 in IMR90 and BJ-hTERT cells as indicated. β-Actin was used as a loading control. One of three independent experimental replicates is shown. **(J)** *CEBPB* mRNA expression was determined by RT-qPCR. One of six independent experimental replicates is shown. Data represent mean ± SD ($n$ = 3). *, P < 0.005 by Student's $t$ test.

**Provided online in separate Excel files are six tables. Table S1 shows the raw data for the epiproteomics analysis in RAS versus control cells. Table S2 lists RAS versus control genes with peaks (FC >1.5) in ChIP-seq and upregulated (FC >1.5; FDR <0.25) in RNA-seq. Table S3 shows the integrated density of cytokines from antibody array. Table S4 lists primers used for the studies. Table S5 shows the annotated ChIP-seq peaks and analysis. Table S6 shows the RNA-seq differential expression analysis.**

