## [Peer Review File · The Journal of Cell Biology]

DOT1L modulates the senescence-associated secretory phenotype through epigenetic regulation of IL1A

Kelly Leon, Raquel Buj, Elizabeth Lesko, Erika Dahl, Chi-Wei Chen, Naveen Tangudu, Yuka Imamura-Kawasawa, Andrew Kossenkov, Ryan Hobbs, and Katherine Aird

Corresponding Author(s): Katherine Aird, University of Pittsburgh School of Medicine

Review Timeline:

Submission Date:	2020-08-21
Editorial Decision:	2020-09-25
Revision Received:	2021-04-06
Editorial Decision:	2021-04-30
Revision Received:	2021-05-05

Monitoring Editor: Ana Pombo

Scientific Editor: Melina Casadio

Transaction Report:

DOI: <https://doi.org/10.1083/jcb.202008101>

September 25, 2020

Re: JCB manuscript #202008101

Dr. Katherine Marie Aird
University of Pittsburgh
5117 Centre Ave
Suite 1.46
Pittsburgh, PA 15213

Dear Dr. Aird,

Thank you for submitting your manuscript entitled "DOT1L modulates the senescence-associated secretory phenotype through epigenetic regulation of IL1A". We sincerely apologize for the delay in sending this decision to you. The manuscript was assessed by expert reviewers, whose comments are appended to this letter. We invite you to submit a revision if you can address the reviewers' key concerns, as outlined here.

You will see that the reviewers - and we agree - found the proposal that, downstream of STING signaling, DOT1L controls IL1a expression and hence influences the SASP very interesting. However, the reviewers were concerned that this model is not adequately supported by the data presented. Both experts were confused by the ability of DOT1L loss to counter the SASP in the presence of high p-p65 levels. One referee suggested that perhaps p-p65 can't fully activate IL1a transcription without DOT1L-mediated chromatin modification. We agree with the reviewers that clarity as to the impact of DOT1L on NF-kappaB signaling and pro-inflammatory cytokine gene control is needed for publication and should be a major focus of the revision. It may be challenging to carry out more refined signaling analyses and experiments modulating IL-1R expression/signaling because most of the signaling apparatus (including the receptor) may be shared between IL-1A and other cytokines (such as IL-1beta). Yet, some of the experiments suggested by the referees (such as Rev#1 #1, overexpression of IL-1A) may help tease out the particular relationship between IL-1A, DOT1L, and NFkB signaling.

The referees additionally suggested adding a more global characterization of the changes in H3K79 methylation upon DOT1L knockdown and overexpression, if not genome-wide at least looking at other SASP loci (Rev#1 #2, #3; Rev#2 paragraph #2), which we agree would help provide valuable genomic context for the results and strengthen your proposal that DOT1L specifically controls the IL1A locus.

Lastly, the reviewers asked about how STING controls DOT1L expression (Reviewer #1 point #4). This is an excellent, relevant question but one that may be better explored in depth in future studies given the scope of this Report. Data addressing this mechanism will not be required for publication. Discussion of this point would be welcome.

The reviewers' constructive feedback about data presentation and discussion and their other points should be carefully considered and addressed.

Please let us know if you have any questions or anticipate any issues addressing these points. We

would be happy to further discuss the reviewers' comments as needed.

GENERAL GUIDELINES:

Text limits: Character count for a Report is < 20,000, not including spaces. Count includes title page, abstract, introduction, results, discussion, acknowledgments, and figure legends. Count does not include materials and methods, references, tables, or supplemental legends.

Figures: Reports may have up to 5 main text figures. To avoid delays in production, figures must be prepared according to the policies outlined in our Instructions to Authors, under Data Presentation, <https://jcb.rupress.org/site/misc/ifora.xhtml>. All figures in accepted manuscripts will be screened prior to publication.

Supplemental information: There are strict limits on the allowable amount of supplemental data. Reports may have up to 3 supplemental figures. Up to 10 supplemental videos or flash animations are allowed. A summary of all supplemental material should appear at the end of the Materials and methods section.

As you may know, the typical timeframe for revisions is three to four months. However, we at JCB realize that the implementation of social distancing and shelter in place measures that limit spread of COVID-19 also pose challenges to scientific researchers. Lab closures especially are preventing scientists from conducting experiments to further their research. Therefore, JCB has waived the revision time limit. Please do not hesitate to reach out to the editors once your lab has reopened to update us about your plans and time frame for resubmission. Please note that papers are generally considered through only one revision cycle, so any revised manuscript will likely be either accepted or rejected.

Thank you for this interesting contribution to Journal of Cell Biology. You can contact us at the journal office with any questions, cellbio@rockefeller.edu or call (212) 327-8588.

Sincerely,

Ana Pombo, PhD
Editor, Journal of Cell Biology

Melina Casadio, PhD

Reviewer #1 (Comments to the Authors (Required)):

In the manuscript by Leon et al. submitted to JCB the authors show that the histone mark H3K79 di- and tri-methylated is enriched during oncogene-induced senescence (OIS). Comparing ChIP-seq and mRNA transcriptomic data, they nicely identify 95 genes which the expression is modulated by this histone modification in OIS, which correspond to genes involved in the pro-inflammatory response and SASP. One of these genes is IL1A, which is SASP master regulator. Then, they show that the histone methyltransferase DOT1L, which is the one responsible for such histone modification is induced in OIS and controls H3K79 methylation and the expression of IL1A and the SASP. Finally, they propose that DOT1L expression is downstream of the cGAS-STING pathway in OIS. In conclusion, they propose that a STING-DOT1L-IL1A pathway controls the SASP. The manuscript is well written, and the experiments are in general well planned and of high quality. The results are clear, and there is no concern about the validity of the role of H3K79 and DOT1L in SASP activation during OIS. This paper is highly relevant as it introduces a new epigenetic layer for the regulation of the SASP. Moreover, the authors show that SASP modulation by DOT1L-H3K79 is independent on the effect on the cell cycle arrest in senescence, which is ideal for serotherapies aiming to modulate SASP effects but not tumour suppressor outcomes. Moreover, the fact that DOT1L has specific inhibitors make this finding more attractive and suggest future exciting research to test this in vivo.

However, the main concern is that the suggested model DOT1L-IL1A-SASP is not robust nor supported by some of the data presented, and other alternative mechanisms are possible. For example, the signal transduction mechanism of IL1A is through IL1R and NF κ B; however, the authors show in figure S2H that targeting DOT1L in OIS does not affect NF κ B signalling (p-p65). This result indicates that lack of DOT1L inhibits SASP induction despite high NF κ B activity, which is at odds with the IL1A-NF κ B-SASP mechanism. A plausible alternative explanation would be that the H3K79me_{2/3} mark is necessary but not sufficient for SASP induction globally. In this model, the H3K79me_{2/3} mark would make the promoters accessible for other transcription factors such as NF κ B or CEBP, and if the mark is absent (e.g. by DOT1L inhibition), the transcriptional activation of the SASP is not possible even with high levels of active RELA (p-p65). Although the results on IL1A H3K79me_{2/3} are clear, the fact that most of the SASP factors (except IL1A) present high basal levels of the H3K79me_{2/3} in control cells which are not increased during OIS, as shown in figure S1H, suggests the interpretation of a global effect on SASP. To clarify this critical point, the authors should address these questions:

- 1- Can IL1A signalling rescue the SASP in OIS in cells lacking DOT1L expression? If IL1A is upstream of the SASP in this context, recombinant active IL1A or overexpression with lentiviral expression vectors should rescue the SASP. What happens to p65 phosphorylation in this context?
- 2- What happens to H3K79 methylation in the other SASP factors (IL1b, il6, il8 etc) when DOT1L is knocked down in OIS? Is DOT1L present in these SASP factors gene loci? If the H3K79me_{2/3} mark is lost in those additional SASP factors, it would suggest a global effect rather than a specific role for IL1A.
- 3- What happens to the H3K79me_{2/3} when DOT1L is overexpressed in the other SASP factors? Do targeting IL1R expression (shRNA) or activity (chemical inhibitors) impair the DOT1L-induced SASP?

Other points.

4- The author suggests that STING is upstream DOT1L expression. STING regulates gene expression by IRF3 or NFkB pathways. How is STING affecting DOT1L expression, is it NFkB dependent or is an interferon response?

5- In the figures, the number of independent experimental replicates and the statistical test used should be indicated. Some of the data is represented as the mean +/- SD. However, in some bar charts, the length of the positive and negative error bar from the mean is different, which is impossible if the data represented is truly the mean +/- SD (e.g. figure 1G, Ras column; figure 1I, RAS column; figure 3C DOT1LOE in DOT1L ChIP etc...). Also, it is not clear if all the bar charts are showing the independent values; this should be specified in the figure legend. Also, I recommend using the standard error instead of the standard deviation if the data represented are truly independent experimental replicates.

If these concerns are addressed, the manuscript shows sufficient quality, novelty and relevance, for publication in JCB, and would be of the interest of readers in the fields of cellular senescence, signal to epigenetics, and cellular biology in general.

Reviewer #2 (Comments to the Authors (Required)):

Leon et al. show that H3K79me2/3 and DOT1L can control inflammatory SASP. The data suggest that H3K79me2/3 is enriched at IL1A, but not IL1B/6/8, loci during senescence. Using RNAi, they show that DOT1L is necessary for induction of IL1a and other inflammatory SASP components. Conversely, OE of DOT1L is sufficient for induction of those cytokines. Finally, they show that STING is required for DOT1L and IL1A induction during OIS.

The potential role of H3K79me2/3 and DOT1L on IL1A expression during senescence is interesting, but the data are largely correlative and not comprehensive. It is difficult to exclude the possibility that DOT1L modulates other gene(s) to regulate the SASP.

One issue is that the functional analysis (RNAi, OE) is only conducted in one single cell line (IMR90 cells). In melanocytes, they only show upregulation of DOT1L and IL1A during OIS. It would be important to repeat the key experiments in this cell type as well.

Page 10. "The decrease in SASP was not due to abrogation of NF- κ B signaling...." As the authors state, IL1A has been proposed to be an upstream effector of the inflammatory SASP. In this model, induction of downstream components of the inflammatory SASP, such as IL1B/6/8, is driven by NFkB (and CEBPb), thus entire signalling is amplified. Then, the lack of abrogation of NF-kB signalling appears inconsistent with their model. More careful characterization of the NF-kB (and possibly CEBPb) pathways would be necessary.

Fig. 3. DOT1L OE is sufficient for the induction of IL1A and other inflammatory SASP components. Since it is an OE condition, they should confirm that k79me2/3 deposition is not increased at the other inflammatory SASP components (i.e. IL1B/6/8). Also, I wonder if they see NF-kB activation.

Minor points

Page 8. "H3K79me2/3 play a role in SASP gene expression during senescence via transcriptional activation of IL1A." This statement sounds a bit too strong. At this stage, it is just a correlation.

p11. "... H3K79me2/3 occupancy at IL1A is necessary for activating SASP gene expression during OIS." This is not supported by the data, just a correlation.

We would like to thank the referees and the Editor for their constructive and thoughtful review of our manuscript submitted as a Report to *Journal of Cell Biology*. We are grateful for their appreciation of the findings that we presented as: “This paper is highly relevant as it introduces a new epigenetic layer for the regulation of the SASP” (Reviewer 1); “The potential role of H3K79me2/3 and DOT1L on IL1A expression during senescence is interesting” (Reviewer 2).

We truly appreciate the importance of getting a set of knowledgeable and helpful reviewers and immensely appreciate and value their effort. Both reviewers raised comments that are truly valuable for the improvement of the manuscript. We believe that by addressing the reviewers' comments and concerns, we have produced a more solid and cohesive manuscript. A point-by-point response to the reviewers' comments and concerns is detailed below with original comments in bold. We hope the Editor and reviewers will find this improved manuscript acceptable.

Reviewer #1

1. Can IL1A signaling rescue the SASP in OIS cells lacking DOT1L expression? If IL1A is upstream of the SASP in this context, recombinant active IL1A or overexpression with lentiviral expression vectors should rescue the SASP. What happens to p65 phosphorylation in this context?

Response: We would like to thank the reviewer for this comment and agree that determining whether IL1A is upstream of the SASP in our model is critical. To address this, we have generated **new data** demonstrating that overexpression of IL1A rescues SASP expression in cells lacking DOT1L expression (**new Fig. 2L-M**).

As recommended by the reviewer, we further looked at the phosphorylation of p65 in this context. Interestingly, overexpression of IL1A in OIS cells lacking DOT1L expression did not affect p65 phosphorylation (**new Fig. S2R**). However, we cannot rule out the possibility that p65 occupancy at these loci is altered, which we have discussed (**Pages**

16-17, Lines 358-367). Additionally, as recommended by Reviewer #2, we also assessed C/EBP β . Our data suggest that C/EBP β expression is altered downstream of the DOT1L-IL1A axis, which may regulate transcription of *IL6*, *IL1B*, and *CXCL8* in our model (**new Fig. 2N and S2R**). We have added discussion on this point (**Page 17, Lines 367-381**).

2. What happens to H3K79 methylation in the other SASP factors (Il1b, il6, il8 etc) when DOT1L is knocked down in OIS? Is DOT1L present in these SASP factors gene loci? If the H3K79me2/3 mark is lost in those additional SASP factors, it would suggest a global effect rather than a specific role for IL1A.

Response: We thank the reviewer for this suggestion. We have generated **new data** to show that knockdown of DOT1L does not globally decrease H3K79 methylation at the *IL6*, *IL1B*, and *CXCL8* loci (**new Fig. S2A-B**), suggesting that H3K79 methylation is playing a specific role for the key SASP regulator IL1A. This is consistent with the idea that H3K79me2 and H3K79me3 occupancy is not increased at the *IL6*, *IL1B*, and *CXCL8* loci during OIS (**new Fig. S1N and Fig. S2A-B**). Together, these data indicate that DOT1L and H3K79 methylation is specific for directly regulating IL1A and not the SASP more globally.

3. What happens to the H3K79me2/3 when DOT1L is overexpressed in the other SASP factors? Do targeting IL1R expression (shRNA) or activity (chemical inhibitors) impair the DOT1L-induced SASP?

Response: We thank the reviewer for these questions. We have generated **new data** to demonstrate that overexpression of DOT1L does not result in the methylation of H3K79me2 or H3K79me3 at the *IL6*, *IL1B*, and *CXCL8* loci (**Fig. S3D-E**). Similar to comment #2 above, these data further support the idea that H3K79 methylation is more specific for the regulation of IL1A.

We additionally thank the reviewer for their question on whether targeting IL1R expression impairs the DOT1L-induced SASP. Although this question is interesting, it

does raise a technical challenge. Modulation of the IL1R may not only affect IL1A, but also other ligands, such as IL1B. Therefore, as suggested by the editor, we did not perform this experiment. We have added a discussion of this limitation to the revised manuscript (**Page 17, Lines 368-371**).

4. The author suggests that STING is upstream DOT1L expression. STING regulates gene expression by IRF3 or NFKB pathways. How is STING affecting DOT1L expression, is it NFKB dependent or is an interferon response?

Response: We would like to thank the reviewer for this excellent question. Addressing this point will be essential towards gaining further understanding of the interaction between STING and DOT1L expression and will allow for elucidation of a novel interaction between the innate response and a methyltransferase. We aim to address this in future studies. As this is a Report, these studies are outside the scope of the current manuscript, which was also pointed out by the Editor. In the revised manuscript, we have discussed how little is known about *DOT1L* regulation at the transcriptional level and pointed out that studies need to be performed to determine whether *DOT1L* upregulation is downstream of NF- κ B or IRF3 (**Page 14-15, Lines 304-316**).

5. In the figures, the number of independent experimental replicates and the statistical test used should be indicated. Some of the data is represented as the mean +/- SD. However, in some bar charts, the length of the positive and negative error bar from the mean is different, which is impossible if the data represented is truly the mean +/- SD (e.g. figure 1G, Ras column; figure 1I, RAS column; figure 3C DOT1LOE in DOT1L ChIP etc...). Also, it is not clear if all the bar charts are showing the independent values; this should be specified in the figure legend. Also, I recommend using the standard error instead of the standard deviation if the data represented are truly independent experimental replicates.

Response: We thank the reviewer for these observations. To improve clarity, we have added additional information about experimental replicates and statistical tests to the

Materials and Methods (**Page 31, Lines 684-689**). We have also added the number of independent experimental replicates performed and the statistical test used for each figure to the legends. We apologize for errors with the length of the positive and negative error bars from the mean. We have double checked to confirm all graphs are correct and additionally added red color to the replicates to allow for better visualization of the data points. As the data shown are technical replicates from independent biological experiments, we have kept standard deviation for figures, unless otherwise indicated.

Reviewer #2

1. The potential role of H3K79me2/3 and DOT1L on IL1A expression during senescence is interesting, but the data are largely correlative and not comprehensive. It is difficult to exclude the possibility that DOT1L modulates other gene(s) to regulate the SASP.

Response: We thank the reviewer for this comment. Similar to our response to Reviewer #1, Comments #2-3, we have generated **new data** to demonstrate that H3K79 methylation at the *IL6*, *IL1B*, and *CXCL8* loci is not globally altered by DOT1L overexpression or knockdown (**new Fig. S2A-B and S3D-E**). These data are consistent with our ChIP-Seq analysis, showing that H3K79me3 is not enriched at other SASP gene loci such as *IL6*, *IL1B*, and *CXCL8* (**Fig. S1N**). Moreover, we have generated **new data** to demonstrate that overexpression of IL1A in DOT1L knockdown cells rescues downstream SASP gene expression (**new Fig. 2L-M**). Together, these data indicate that DOT1L does not directly affect other SASP genes or other genes that regulate SASP expression, but instead promotes the SASP through regulating *IL1A*.

2. One issue is that the functional analysis (RNAi, OE) is only conducted in one single cell line (IMR90 cells). In melanocytes, they only show upregulation of DOT1L and IL1A during OIS. It would be important to repeat the key experiments in this cell type as well.

Response: We thank the reviewer for this suggestion. We agree that it is important to repeat key experiments in a second model. We are unable to perform key experiments such as ChIP in melanocytes as their growth rate is prohibitive for more technically-challenging experiments. To address the reviewer's comment, we have performed experiments in a second commonly-used model of oncogene-induced senescence: expression of BRAF^{V600E} in BJ-hTERT human foreskin fibroblasts (**new Fig. S1G-L**). These data demonstrate an increase in DOT1L expression (**new Fig. S1T and S1W**) and H3K79me2/3 methylation at the *IL1A* locus in BRAF^{V600E}-induced senescence (**new Fig. S1P-Q**), correlating with *IL1A* expression (**new Fig. S1R**). Through knockdown and overexpression studies, we further confirm that DOT1L is both necessary and sufficient for SASP gene expression in this model (**new Fig. S2H-I, S3B-C, and S3F-G**) without altering other senescence phenotypes (**new Fig. S2J-L and S3K-L**). These data indicate that the observed affects are not cell line- or oncogene-specific.

3. Page 10. "The decrease in SASP was not due to abrogation of NF-κβ signaling...."
As the authors state, IL1A has been proposed to be an upstream effector of the inflammatory SASP. In this model, induction of downstream components of the inflammatory SASP, such as IL1B/6/8, is driven by NFκB (and CEBPb), thus entire signalling is amplified. Then, the lack of abrogation of NF-kB signalling appears inconsistent with their model. More careful characterization of the NF-kB (and possibly CEBPb) pathways would be necessary.

Response: We thank the reviewer for this comment. Indeed, we agree with the reviewer that NF-κβ helps to amplify SASP signaling downstream of IL1A. Therefore, we have removed that statement. A previous study from the Campisi lab demonstrated that inhibition of IL1A signaling decreases p65 DNA binding (Orjalo et al., 2009). It is possible that while p65 phosphorylation is maintained, its nuclear translocation, DNA binding ability, or full transcriptional activation is somehow inhibited in cells with DOT1L knockdown. We have added this as a potential caveat to our studies and a discussion of this important point to the revised manuscript (**Pages 16-17, Lines 358-367**).

We also agree with the reviewer that further determining whether C/EBP β plays a role in regulating the expression of SASP is essential towards better understanding the mechanisms downstream of DOT1L. Knockdown of DOT1L decreased C/EBP β expression, and overexpression of IL1A in DOT1L knockdown cells partially rescued its expression, correlating with increased *IL6*, *IL1B*, and *CXCL8* (**new Fig. 2L-N and S2R**). These data suggest that changes in C/EBP β expression are downstream of IL1A, which is consistent with previous reports showing that knockdown or activation of IL1A positively correlates with C/EBP β expression (Hop et al., 2019; Montes et al., 2015; Yang et al., 2015). Moreover, H3K79me₃ ChIP-Seq analysis demonstrates that there is no enrichment of H3K79me₃ at the *CEBPB* locus (**new Fig. S2S**), further suggesting that *CEBPB* is not a direct target of DOT1L but likely downstream of changes in IL1A. Together, we propose a model in which DOT1L promotes IL1A expression to amplify SASP gene expression in part via C/EBP β . As both C/EBP β and NF- κ B are critically important transcriptional regulators of the SASP, future work will be aimed at better understanding the interplay between C/EBP β and NF- κ B signaling downstream of DOT1L. We have added discussion of these points to the revised manuscript (**Pages 16-17, Lines 354-381**).

4. Fig. 3. DOT1L OE is sufficient for the induction of IL1A and other inflammatory SASP components. Since it is an OE condition, they should confirm that k79me_{2/3} deposition is not increased at the other inflammatory SASP components (i.e. IL1B/6/8). Also, I wonder if they see NF- κ B activation.

Response: We greatly appreciate the reviewer's comments. We have **new data** to demonstrate that occupancy of H3K79me_{2/3} is not increased at *IL6*, *IL1B*, or *CXCL8* (**new Fig. S3D-E**). These data further support the notion that DOT1L does not directly regulate the expression of *IL6*, *IL1B*, or *CXCL8*. Additionally, overexpression of DOT1L does not alter phosphorylation of p65 (**new Fig. S3T**), while we did observe an increase in C/EBP β expression (**new Fig. S3U-V**). It is important to note that overexpression of DOT1L does not increase SASP gene expression to the same extent as in oncogene-induced senescent cells, suggesting that the lack of p65 phosphorylation in DOT1L

overexpressing cells may blunt the full transcription of SASP genes. Therefore, the data indicate that while DOT1L is necessary for SASP gene expression, it likely also requires induction of other factors (in particular phosphorylation and DNA binding of p65) to fully promote expression of downstream SASP genes. We have added a discussion of this point to the revised manuscript (**Page 15, Lines 322-326**).

5. Page 8. "H3K79me2/3 play a role in SASP gene expression during senescence via transcriptional activation of IL1A. " This statement sounds a bit too strong. At this stage, it is just a correlation.

Response: We thank the reviewer for this comment and have changed the sentence to read: "Together, these data suggest that H3K79me2/3 may play a potential role in SASP gene expression during senescence via transcriptional activation of *IL1A*."

6. p11. ".... H3K79me2/3 occupancy at IL1A is necessary for activating SASP gene expression during OIS. " This is not supported by the data, just a correlation.

Response: We thank the reviewer for this comment. We agree that the statement is too strong, and we have therefore removed it from the revised manuscript.

References

- Hop, H.T., A.W.B. Reyes, L.T. Arayan, T.X.N. Huy, S.H. Vu, W. Min, H.J. Lee, C.K. Kang, M.H. Rhee, and S. Kim. 2019. Interleukin 1 alpha (IL-1 α) restricts *Brucella abortus* 544 survival through promoting lysosomal-mediated killing and NO production in macrophages. *Vet Microbiol.* 232:128-136.
- Montes, M., M.M. Nielsen, G. Maglieri, A. Jacobsen, J. Højfeldt, S. Agrawal-Singh, K. Hansen, K. Helin, H.J.G. van de Werken, J.S. Pedersen, and A.H. Lund. 2015. The lncRNA MIR31HG regulates p16(INK4A) expression to modulate senescence. *Nature communications.* 6:6967.
- Orjalo, A.V., D. Bhaumik, B.K. Gengler, G.K. Scott, and J. Campisi. 2009. Cell surface-bound IL-1alpha is an upstream regulator of the senescence-associated IL-6/IL-8 cytokine network. *Proceedings of the National Academy of Sciences of the United States of America.* 106:17031-17036.
- Yang, B., W. Li, Q. Zheng, T. Qin, K. Wang, J. Li, B. Guo, Q. Yu, Y. Wu, Y. Gao, X. Cheng, S. Hu, S.N. Kumar, S. Liu, and Z. Song. 2015. Transforming growth factor β -activated kinase 1 negatively regulates interleukin-1 α -induced stromal-derived factor-1 expression in vascular smooth muscle cells. *Biochemical and biophysical research communications.* 463:130-136.

April 30, 2021

RE: JCB Manuscript #202008101R

Dr. Katherine Marie Aird
University of Pittsburgh School of Medicine
5117 Centre Ave
Suite 1.46
Pittsburgh, PA 15213

Dear Dr. Aird,

Thank you for submitting your revised manuscript entitled "DOT1L modulates the senescence-associated secretory phenotype through epigenetic regulation of IL1A". Thank you very much for your patience with the re-review process. As Ana Pombo was not available, JCB Editor-in-Chief Jodi Nunnari is handling the revision with me (M.C.). We and the reviewers are enthusiastic about the revision and commend you for your revision efforts to address the referees' concerns. The model is stronger and we would be very happy to publish your paper in JCB pending final revisions necessary to meet our formatting guidelines (see details below). Please also consider Rev#1's final point as you prepare your final files.

- 1) JCB Reports must have a combined "Results and Discussion" section. Could you please reword the "Results" header and remove the "Discussion" header? Please let me know if you have any questions about this change.
- 2) JCB Reports can have up to 5 main and 4 supp figures. Would you be willing to move some of the supp data to the main figures and/or or to rearrange it please? Each figure can span up to 1 entire page as long as all panels fit on the page.
- 3) Figure formatting:
Molecular weight or nucleic acid size markers must be included on all gel electrophoresis. Please add molecular weight with unit labels on the following panels: 1CK, 2AL, 3AB, 4BD, S1BGLW, S2CHMNR, S3TCHBV, S4A
- 4) Statistical analysis: Error bars on graphic representations of numerical data must be clearly described in the figure legend. The number of independent data points (n) represented in a graph must be indicated in the legend. Statistical methods should be explained in full in the materials and methods. For figures presenting pooled data the statistical measure should be defined in the figure legends.
- 5) Materials and methods: Should be comprehensive and not simply reference a previous publication for details on how an experiment was performed. Please provide full descriptions in the text for readers who may not have access to referenced manuscripts.
- For all cell lines, vectors, constructs/cDNAs, etc. - all genetic material: please include database /

vendor ID (e.g., Addgene, ATCC, etc.) or if unavailable, please briefly describe their basic genetic features *even if described in other published work or gifted to you by other investigators*

- Please include species and source for all antibodies, including secondary, as well as catalog numbers/vendor identifiers if available.
- Sequences should be provided for all oligos: primers, si/shRNA, gRNAs, etc.
- Microscope image acquisition: The following information must be provided about the acquisition and processing of images:
 - a. Make and model of microscope
 - b. Type, magnification, and numerical aperture of the objective lenses
 - c. Temperature
 - d. imaging medium
 - e. Fluorochromes
 - f. Camera make and model
 - g. Acquisition software
 - h. Any software used for image processing subsequent to data acquisition. Please include details and types of operations involved (e.g., type of deconvolution, 3D reconstitutions, surface or volume rendering, gamma adjustments, etc.).

6) References: There is no limit to the number of references cited in a manuscript. References should be cited parenthetically in the text by author and year of publication. Abbreviate the names of journals according to PubMed.

- Please be sure to format preprint citations (in the text and in the ref list) as per our guidelines: <https://rupress.org/jcb/pages/reference-guidelines>

"Buj, R., Leon, K.E., and Aird, K.M. (2020). Suppression of p16 alleviates the senescence-associated secretory phenotype. bioRxiv"

A. MANUSCRIPT ORGANIZATION AND FORMATTING:

Full guidelines are available on our Instructions for Authors page, <https://jcb.rupress.org/submission-guidelines#revised>. **Submission of a paper that does not conform to JCB guidelines will delay the acceptance of your manuscript.**

B. FINAL FILES:

****It is JCB policy that if requested, original data images must be made available to the editors. Failure to provide original images upon request will result in unavoidable delays in publication. Please ensure that you have access to all original data images prior to final submission.****

****The license to publish form must be signed before your manuscript can be sent to production. A link to the electronic license to publish form will be sent to the corresponding author only. Please take a moment to check your funder requirements before choosing the appropriate license.****

Thank you for this interesting contribution, we look forward to publishing your paper in Journal of Cell Biology.

Sincerely,

Jodi Nunnari, Ph.D.
Editor-in-Chief, Journal of Cell Biology

Melina Casadio, Ph.D.
Senior Scientific Editor, Journal of Cell Biology

Reviewer #1 (Comments to the Authors (Required)):

The authors have addressed my comments satisfactorily.
My only question is if in figure 1B, performing student's t-test with two replicates meets the standards of JCB. I would recommend not to show the statistical test in this case.
Otherwise, I recommend the publication of the manuscript.

Reviewer #2 (Comments to the Authors (Required)):

the authors have added a substantial amount of new data, satisfactorily addressing reviewers' questions.